# Inventory mapping of forest-covered landslides using Geographic Object-Based Image Analysis (GEOBIA), Jena region, Germany

Ikram Zangana[1], Rainer Bell[1], Lucian Drăguţ[2], Flavius Sîrbu[3] and Lothar Schrott[1]

[1]Department of Geography, University of Bonn, Bonn, Germany
[2]Department of Geography, West University of Timişoara, Timişoara, Romania
[3]Institute for advanced environmental research, West University of Timişoara, Timişoara, Romania

*Correspondence to*: Ikram Zangana (ikram.zangana@uni-bonn.de)

**Abstract.** Landslide inventories are crucial for the assessment of landslide susceptibility and hazard. An analysis of old landslides can reveal periods of intensified landslide activity, but the features of these landslides may have diminished over time, particularly in the context of human impact. However, landslide features are often preserved well under forest cover and are thus valuable for compiling or updating landslide inventories. However, the mapping of these features remains challenging. Light detection and ranging (lidar) analysis and its derivatives are essential in landslide research, particularly in landslide

identification and mapping. Unlike the expert-based analysis of lidar derivatives, the use of object-based approaches to map landslides from lidar data (semi)automatically requires further studies. This study adopts geographic-object-based image analysis based solely on lidar derivatives for the inventory mapping of forest-covered old landslides within a middle-mountain region in Jena, Germany, and surrounding areas. A manually prepared expert-based inventory map was used for model training and validation. Lidar derivative data were processed using (a) a default moving-window size (3 × 3; model I) and (b) an optimal

window size (model II). Multi-resolution segmentation and support vector machine classification with distinct rule sets were implemented for each model, followed by refinement and accuracy assessment against the inventory map for model performance evaluation. The proposed approach achieved a 70% detection of existing landslides compared with the inventory. Model II outperforms model I in accuracy, as indicated by its superior performance in scarp area detection (15% improvement) and significantly lower false positives (30% reduction). However, although this method excellently identifies and maps forest-

covered old landslides, its applicability is currently limited to large and medium landslides (area > 0.5 ha). Overall, our findings suggest that landslides worldwide with clear geomorphological signatures in lidar data can be identified using this approach.

## 1 Introduction

Landslides are significant in landform evolution, and numerous regions worldwide have considerable landslide hazard. In certain areas, landslides frequently cause greater mortality and economic loss than other natural hazards, such as earthquakes,

volcanic eruptions and floods (Guzzetti et al., 1999; Aksoy and Ercanoglu, 2012; Guzzetti et al., 2021). Landslide hazard is

the probability of a landslide of a specific magnitude occurring in a particular area within a defined time frame (Guzzetti et al., 1999). Landslide hazard assessment necessitates the creation of detailed landslide maps, with landslide inventory maps specifically recording the geographic distribution of documented landslides based on their detection and delineation (Guzzetti et al., 1999). Such inventories are traditionally developed through analyses of aerial photographs, supplemented by fieldwork and collection of historical data (Guzzetti et al., 1999, 2012; Santangelo et al., 2010). Albeit a standard geomorphological practice, the field mapping of landslides, particularly older ones, is hindered by factors such as landslide size, limitations in field perspectives, forest cover or erosion and anthropogenic modifications. Compared with traditional field techniques, remote methods using aerial photographs and high-resolution digital elevation models (DEMs) provide more comprehensive and accurate data, increasing mapping precision (Santangelo et al., 2010; Guzzetti et al., 2012; Bell et al., 2012; Crawford, 2014; Schmaltz et al., 2016; Petschko, Bell, and Glade, 2016; Bernat Gazibara et al., 2019).

In recent decades, geographic-object-based image analysis (GEOBIA) has emerged as a powerful method for semi- or fully automated landform mapping (e.g. Drăguţ and Blaschke, 2006; Blaschke, 2010a; Schneevoigt et al., 2010; Seijmonsbergen et al., 2011; Anders et al., 2011, 2013; Drăguţ and Eisank, 2012; Zylshal et al., 2013; Eisank et al., 2014; Robb et al., 2015; Pedersen, 2016; Guilbert and Moulin, 2017; Hossain and Chen, 2019). The integration of GEOBIA into semi-automatic landslide mapping is a significant development in this field. Lahousse et al. (2011) developed a multiscale GEOBIA technique for landslide mapping, but it is limited to specific areas and landslide types. Aksoy and Ercanoglu (2012) proposed a semi-automatic inventory mapping method that uses fuzzy logic based on thematic data and spectral information. Feizizadeh and Blaschke (2013) developed a rule-based classification approach utilising satellite data. Hölbling et al. (2016) identified spatiotemporal landslide hotspots by analysing historical and recent aerial photographs. Hölbling et al. (2017) compared GEOBIA and manual mapping approaches and concluded that GEOBIA-based semi-automatic mapping encounters difficulties in areas where landslides are covered by vegetation. Karantanellis et al. (2020, 2021) stated that landslide modelling based on unmanned aerial vehicles (UAVs) enables detailed, automated landslide characterisation, with high adaptability to specific sites. They also found that UAVs enable time- and cost-efficient data collection, whereas machine learning algorithms are effective for local-scale sub-zone landslide mapping when integrated into GEOBIA. Dias et al. (2023) showed that applying GEOBIA-based methods to high-resolution satellite imagery can successfully identify shallow landslides and debris flows with over 70% accuracy. Karantanellis and Hölbling (2025) further emphasised the utility of high-resolution digital data, in combination with GEOBIA-based methods, for improving landslide mapping and assessment accuracy.

Limited studies have explored the use of GEOBIA for landslide mapping in forested regions, particularly in the underexamined context of old landslide inventories. Plank and Martinis (2016) used an object-based and change detection approach with DEM and synthetic-aperture radar (SAR) imagery to map landslides in vegetated areas by integrating pre-event optical and post-event very-high-resolution polarimetric SAR data. However, their study focused only on fresh landslides, not old landslides under forest cover. Comprehensive inventory mapping is required to address this limitation. Eeckhaut et al. (2006, 2012) studied landslides occurring beneath forest cover, achieving a detection rate of approximately 70% using lidar data alone. Their investigation encompassed multiple levels, but the moving-window size of land surface variables (LSVs) for

landslide components (e.g. landslide scarp and body) were not adequately addressed; both components were treated using the same window size. Knevels et al. (2019) used open-source software to map forest landslides using GEOBIA. Using high-resolution lidar data, they attained a 69% detection rate relative to manual mapping. Despite using default window sizes, the authors acknowledged the potential of identifying the optimal window size for different landslide portions, possibly enhancing model performance. In summary, previous researchers used default window sizes to calculate LSVs, underscoring the need to

explore optimal moving-window approaches to enhance landslide mapping.

   Determining the optimal window sizes for different LSVs relative to specific landforms is critical in semi-automatic landform detection using digital data; a detailed review of scale-related issues is available in Drăguţ and Eisank (2011). Seijmonsbergen et al. (2011) demonstrated that using multiple window sizes for LSVs can enhance semi-automatic landform detection. They found that different landscape features are best detected using different window sizes, but they manually

selected these sizes to compare with expert-based mapped features. Pawluszek et al. (2018) investigated the impact of scaling window sizes on the automatic detection of landslides using digital terrain model (DTM) data. After DTM rescaling, the landslide modelling accuracy improved relative to that of the original (non-rescaled) DTM. Sîrbu et al. (2019) developed an automated approach to selecting the optimal window size of each LSV relative to landslide scarps, significantly improving detection accuracy in two study sites in comparison with that under the default selection of window sizes. However, landslide

bodies were not examined in this study. No standard or operational method has been developed to achieve this goal despite the considerable progress in automated landslide mapping.

   This study investigates the potential of using GEOBIA and high-resolution DTM data for the semi-automatic mapping of forest-covered old landslides (mainly focus on the deep-seated 'rotational' landslides) in middle-mountain regions in Jena, Germany. Specifically, the effectiveness of using lidar data and their derivatives for the semi-automated inventory mapping of

forest-covered landslides is assessed, particularly the role of optimised window sizes. The central research question is as follows: How can DTM derivatives and optimised window sizes enhance the reliability of GEOBIA-based semi-automatic landslide mapping in forested environments? Thus, the influence of optimal window sizes for LSVs on the accuracy of semi-automatic landslide mapping is first determined. This is then compared with results achieved using default window sizes. By addressing these aspects, this study seeks to advance the understanding of, and improve practices in, landslide mapping within

forested environments.

## 2 Study area

   The study area is in the eastern part of Thuringia, near the city of Jena, Germany (Figure 1). It is approximately 150 km$^2$ in size and encompasses two elevation zones. The first zone is a low-elevation area that includes most of the Saale River valley and parts of the Roda River catchment. The other is an elevated zone consisting of a plateau, low mountains and adjacent

slopes at elevations reaching 400 m asl (Zangana et al., 2023). Moreover, the study area is situated within the Thuringian Basin and has two predominant geological formations. The Muschelkalk Formation (limestone) predominates in the higher-altitude

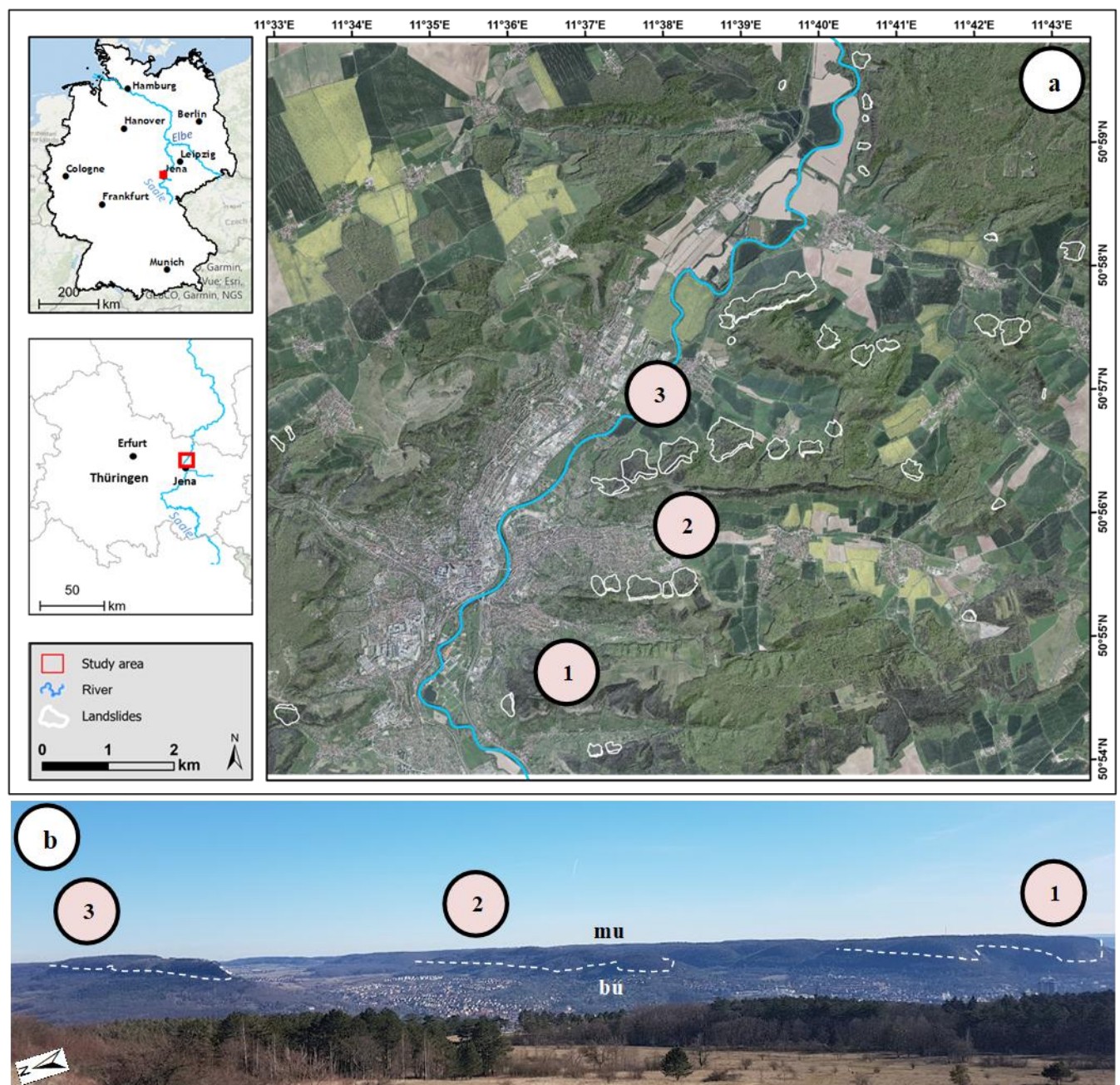

**Figure 1.** Study area. (a) Map: An orthophoto of the study area is overlaid on a hillshade DTM, with the landslide inventories delineated by white polygons. (b) Photo (taken by Ikram Zangana on February 24, 2019): The photo faces the southeast direction, showing the hillslopes of Kerenberge, Hausberg and Jenzig, along with a forested area and part of Jena City. Key locations are marked (1, 2 and 3), and the white dashed lines indicate the lithological boundaries, with Muschelkalk (mu) limestone above and Buntsandstein (bu) sandstone below. The orthophoto and the hillshade-DTM in (a) were obtained from TLUBN (2019) and other datasets are sourced from ESRI (2025).

regions, whereas the Buntsandstein Formation (red sandstone) dominates in the lower-altitude areas (Föhlisch, 2002; Seidel, 1992).

The preconditioning factors influencing landslide occurrence in the study area are geological and structural characteristics, particularly the stratigraphic contact between limestone and underlying sandstone. This lithological configuration, when combined with the steep slope geometry of the cuesta escarpments, plays a pivotal role in the slope instability. Such structural settings are known to favour rotational landslides, particularly in cuesta landscapes, where differential weathering and erosion of layered sedimentary rocks promote mass movement. According to Achilles et al. (2016),
the landslides may have been triggered during the Holocene, likely beginning at the end of the Weichselian glaciation, due to increased precipitation, glacial meltwater infiltration, and associated hydrological changes. While the exact age of these landslides remains uncertain, especially when relying solely on LiDAR-derived DTMs, the widespread presence of dense forest cover over many large landslide bodies suggests limited recent activity and supports the possibility of an older origin (Zangana et al., 2024). Most of the mapped landslides occur on hillslopes in the eastern part of the Saale River basin,
predominantly facing north and northwest. However, to the best of our knowledge, there are no recent studies or official records documenting damage or economic losses correlated with older deep-seated landslides in the region.

    The annual mean temperature is 9°C–11°C, the summer mean is approximately 16°C–18°C and the winter mean is 0°C–2°C. The mean annual rainfall is 600–800 mm (TMUEN, 2017). Land use is dominated by residences, industries and infrastructure in the valley floors and some gentle slopes. Forests cover steep slopes and high plateaus. Farmlands are primarily
located along the floodplains of the Saale River and its tributaries, whereas grasslands and pastures are more sparsely distributed, mainly in the northern portion of the study area (landnutzung). The soil types in this area include rendzinas (Leptosols), which are on the Muschelkalk Formation, predominantly within the plateau area, and pararendzinas (Pelosols), which are in the Buntsandstein area and on the slopes. However, the Holocene floodplain and flat areas of the region are covered by gley–Vega soil types (Gleysols). Cambisols are found in areas dominated by sandstone, sandstone/siltstone and
claystone sequences of the lower and middle Buntsandstein, while podsols (Podzols) are present in some southern parts of the study area (Rau et al., 2000; Zangana et al., 2023b).

## 3 Methodology

### 3.1 Data

Landslide mapping is based on lidar–DTM data with a 1 m × 1 m resolution provided by the Thuringian State Office for Soil
Management and Geoinformation (TLUBN, 2019; Zangana et al., 2023a). Different LSVs, namely, slope, topographic openness (TO), curvature (plan and profile), terrain roughness index (TRI) and topographic position index (TPI), were generated using an original DTM. A landslide inventory map (reference map) was created using manual on-screen mapping in ArcMap 10.7. Traditional and multi-directional hillshade were used as the primary visual base, following the method described by Schulz (2004). Hillshades and slope maps were visually evaluated for landslide features such as scarps and bodies

by systematically panning through the imagery at scales ranging from 1:1,000 to 1:200 and mapped accordingly at this scale to ensure the correct delineation of landslide boundaries. However, additional LSVs (e.g. curvature, TO, TPI, and TRI) were employed as supplementary layers to facilitate interpretation and boundary delineation, particularly in regions where hillshade and slope alone were inadequate for fully resolving the geomorphic manifestation of landslides. Furthermore, as in the method applied by Zangana et al. (2023b), we incorporated a LSVs-composite map visualisation that improved the detection of

morphological features of landslides. Scarps and bodies were mapped separately wherever they could be clearly distinguished. In a few instances, scarp features could not be identified with confidence from the available data. Approximately 10% of the mapped landslides were validated in the field. The inventory primarily includes deep-seated (rotational) landslides (34 landslides), along with a few shallow landslides (6 landslides).

### 3.2 GEOBIA-based landslide inventory mapping

We used the software eCognition 10.3 and developed a structured workflow to design a rule set for semi-automatic landslide mapping. This workflow enabled landslide identification using two distinct models. Model I (MI) used the default window size to calculate the LSVs as a pre-processing step for segmentation and classification, whereas model II (MII) used the optimal window size. The final results were exported as shapefiles to ArcGIS 10.7. The overall methodological framework, consisting of three main stages, is illustrated in Figure 2.

**3.2.1 STAGE I: Data preparation**

This stage was divided into two main steps. Step 1: A landslide inventory map was manually prepared using DTM hillshade data and a visual analysis of all relevant LSVs. This inventory map served as a reference for model development and as a baseline with which the final GEOBIA results were compared for accuracy assessment (AA). Step 2: ArcGIS 10.7 and R 4.3.2 were used to generate LSVs using different window sizes. For MI, the default window sizes in ArcGIS were applied based on

standard raster calculation methods and commonly published values. For MII, we adopted an advanced approach involving the automatic detection of optimal window sizes for each LSV for alignment between landslide-prone and non-landslide areas (Sîrbu et al., 2019).

      For model training samples were collected from landslide scarps (MI: 6.09 ha; MII: 1.32), landslide bodies (MI: 36.20 ha; MII: 27.2 ha) non-scarp areas (MI: 9.42 ha; MII: 4.07 ha), non-body areas (MI: 53.95 ha; MII: 41.21 ha). These values

represent the total sampled area used for each model (MI and MII). Classification was then performed using unsupervised methods and support vector machines (SVMs; for further details, see Tzotsos and Argialas, 2008; Hong et al., 2017). An algorithm was trained using these samples and then used to classify the data accordingly. For optimal results, this stage was repeated multiple times while adjusting the training samples iteratively to enhance accuracy in comparison with the inventory map.

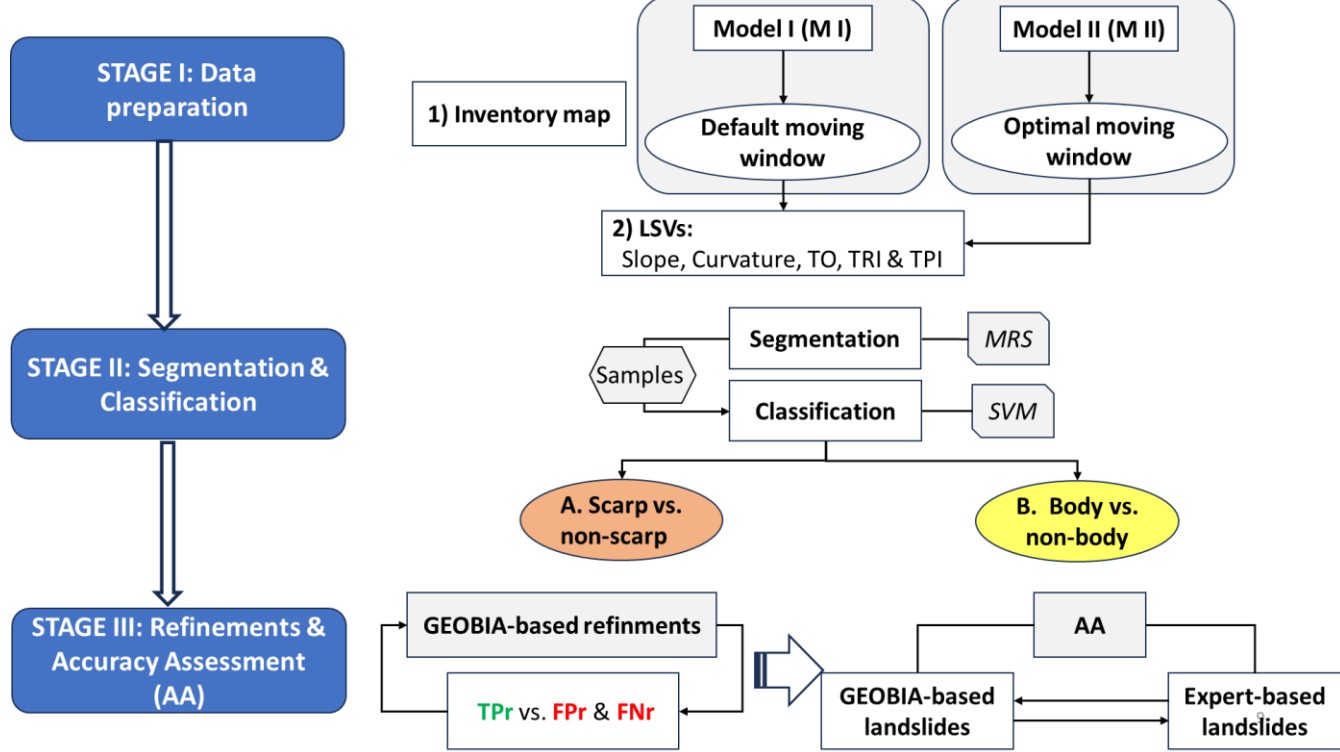

**Figure 2.** Flowchart of method used for mapping of landslide scarps and bodies. First, data are prepared using default and optimal window sizes. Then, segmentation and classification are conducted via MRS and SVM. Finally, refinement and accuracy assessment (AA) are performed by comparing the reference map with the GEOBIA results.

### 3.2.2 STAGE II: Segmentation and classification

Segmentation and classification were performed using eCognition at two hierarchical levels. Segmentation was conducted using multi-resolution segmentation (MRS; Baatz and Schäpe, 2000) after trial-and-error across different scales (Drăguţ et al., 2010; Li et al., 2015) for distinct landslide components (landslide scarps and bodies). Specifically, landslide scarps and bodies were segmented and identified using separate projects and rule sets. MRS with scale parameters of 50 and 20 for landslide scarps using shape of 0.1 and compactness of 0.5 and MRS with scale parameters of 70 and 30 for landslide bodies using shape of 0.1 and compactness of 0.5 achieved the best fit and were thus applied to MI and MII, respectively.

### 3.2.3 STAGE III: Refinement and accuracy assessment (AA)

This stage consisted of two main steps: GEOBIA-based refinement and accuracy assessment (AA). The first, a GEOBIA-based refinement aimed to enhance the initial SVM classification from Stage II by incorporating additional object-based rules. Based on the outcomes of Stage II, an additional stage (Stage III) was developed using expert knowledge and implemented as a rule

set within the eCognition framework. This GEOBIA-based refinement leverages expert-driven interpretation in combination with object-level spatial and contextual information to enhance classification accuracy. Previous studies have demonstrated the value of combining machine learning with rule-based approaches for improving thematic mapping quality (Johnson and Xie, 2011; Eisank et al., 2014;  Zylshal et al., 2016; Robson et al., 2020).

The refinement process focused on iteratively improving the true positive rate (TPr) while reducing both the false positive rate (FPr) and false negative rate (FNr). This was achieved by assessing a comprehensive set of object-based features derived from morphometric (e.g., slope, curvature, TPI, ...etc.), geometric (e.g., area, shape index, length-to-width ratio), and contextual attributes (e.g., distance to landslide-related objects, and the relative border to neighbor metric). The latter measures the proportion of an object's boundary shared with a predefined class, helping to identify embedded or adjacent features for
reclassification. For example, a relative border value of 1 indicates complete enclosure by a reference class, while lower values suggest partial adjacency (for a further and comprehensive overview of the ruleset developed, see Tables A1-B2 in the Appendices).

Refinements were conducted using eCognition's interactive visualization tools, enabling semi-automated object filtering and targeted adjustments based on spatial inconsistencies. Objects were assessed through iterative cycles of visual
inspection, attribute filtering, and validation against the reference inventory using both number- and area-based accuracy metrics. Key decisions included merging or expanding TP-classified regions and reclassifying ambiguous objects based on rules such as (1) adjacency to existing TP objects, (2) sharing >80% boundary with TPs, and (3) being fully surrounded by TP zones. Importantly, no expansion was allowed into areas clearly identified as non-landslide terrain. This process was repeated until no further improvements were observed. The landslide scarps and bodies were analysed separately, so different criteria
and parameters were applied to each of them during the development of the rule set in eCognition. In other words, the landslide scarp area was treated separately from the landslide body area.

The second step, called AA, involved comparing the final result of GEOBIA-based refinement (stage III, part 1) with expert-based landslide data (i.e. the inventory map). This comparison helped assess the efficacy of each model (MI and MII) against the reference map. For a comprehensive investigation, AA was conducted separately for landslide scarps and bodies.
As seen in Section 3.3, number-based AA, area-based AA and calculation of additional metrics were adopted (Cai et al., 2018; Simoes et al., 2023).

**3.3 Accuracy assessment**

Various metrics were used to evaluate the congruence between the GEOBIA mapping outcomes and the reference map quantitatively. These comparisons were conducted independently per model and per landslide component: landslides scarps
and bodies.

### 3.3.1 Thematic accuracy assessment

We assessed the results through number-based and area-based accuracy assessment. First, following Cai et al. (2018), we developed an R script to assess the accuracy of the model results numerically. If the GEOBIA-detected polygon's overlapping area exceeded 50% of the area of the reference landslide polygon, then it was considered a correctly identified landslide (Eeckhaut et al., 2012; Knevels et al., 2019). The TP, FP and FN numbers and percentages were calculated according to MI and MII. Then, in addition to number-based AA, area-based AA (hectares [ha]) was adopted to obtain more detailed information about the absolute areas correctly detected as landslides (i.e. TP), undetected landslide areas (i.e. FN) and areas incorrectly mapped as landslides (i.e. FP). To achieve this, we overlaid the inventory map polygons (reference map) on the GEOBIA-based polygons, which included these three components, to calculate the percentage of each category and determine whether the use of the optimal moving-window size in MII improved the semi-automatic GEOBIA-based mapping results (Figure 6). The script for this analysis was developed and implemented in R using the GEOBIA results according to previously reported key concepts and algorithms (for further details, see Eisank et al., 2014; Cai et al., 2018).

### 3.3.2 Segmentation metrics

An R script developed using the *segmetric* package to calculate the segmentation accuracy of the objects of interest through various metrics (Simoes et al., 2023). As outlined in Table 6, we analysed key metrics relevant to landslide studies: area fit index (AFI), over-segmentation (OS), under-segmentation (US), F-measure, recall and precision. These metrics were based on area proportions, with values between 0 and 1 except for AFI. A value closer to zero indicated a better spatial match between the test and reference datasets (Dias et al., 2023).

## 4 Results

This chapter presents the main results of this study, specifically the optimisation results of window sizes for LSVs in MII, the results of GEOBIA-based landslide detection and the model performance evaluation results from different AA approaches.

### 4.1 Optimisation of window sizes for LSVs in MII

Figure 3 shows the variation in the optimal window size for each LSV across multiple runs, highlighting noticeable differences between landslide scarps and bodies. Some LSVs (e.g. TO, plan curvature and profile curvature) have consistent window sizes, whereas others (e.g. slope and TRI) show greater variability. Table 1 shows the final window sizes used in the analysis for both landslide components in MII, along with the default values used in MI for comparison. The optimal window sizes differ not only between LSVs but also between scarps and bodies within the same LSV. Hence, separate rule sets were developed in eCognition for each landslide component. Segmentation, classification, refinement and AA were then performed independently for the scarps and bodies.

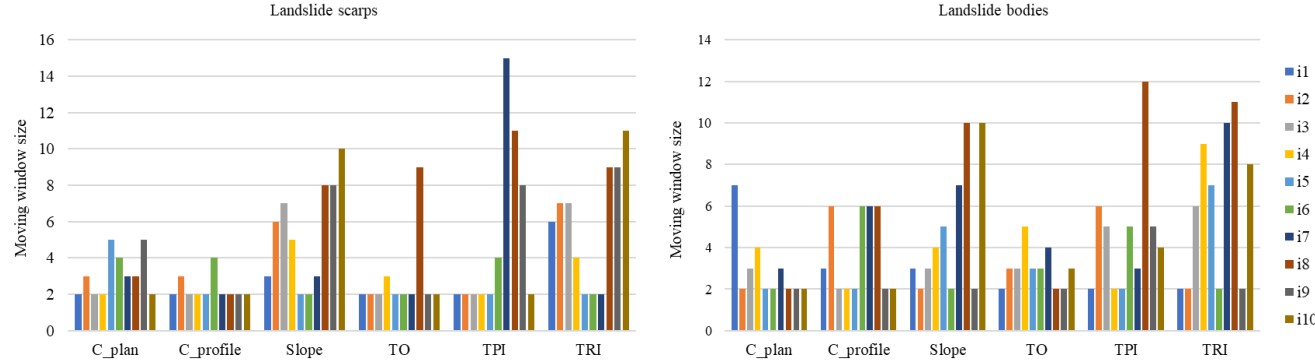

**Figure 3.** Optimal window sizes for each LSV in predicting landslide scarps and bodies. Note: i1, i2, ..., i10: number of iterations; C: curvature; TO: topographic openness; TRI: terrain roughness index; TPI: terrain position index.

**Table 1**. LSVs and corresponding window sizes per model.

| Models/LSVs | | Slope | C_plan | C_profile | TO | TRI | TPI |
|---|---|---|---|---|---|---|---|
| I * | Scarp & Body | 3x3 | 3x3 | - | 3x3 & 25x25 | 3x3 | 33x33 |
| II ** | Scarp | 11x11 | 7x7 | 5x5 | 5x5 | 7x7 | 11x11 |
| | Body | 11x11 | 5x5 | 7x7 | 7x7 | 9x9 | 13x13 |

\* Default window size, \*\* optimal window size

## 4.2 GEOBIA-based landslide modelling results

The GEOBIA-based landslide modelling results, specifically those of landslide scarps and bodies for both models (MI and MII), were compared with the inventory map to assess their spatial correspondence (Figures 4 and 5). Figure 4(a) shows the MI results (default window sizes). The brown and yellow polygons represent the model-detected landslide scarps and bodies, and the blue and pink polygons indicate the landslide scarps and bodies in the reference map for comparison. Figure 5(a) shows the same area but illustrates the MII results. In both figures, the polygons within the black-dashed-line regions are further discussed in Section 5. A visual inspection of these maps shows that MI covers a larger portion of the landslide body areas compared with MII, but MII performs better in detecting scarp zones. However, on-screen analysis shows that MII is more precise than MI for each landslide component, as indicated by the brown and yellow polygons (GEOBIA-based results) and

the blue-(scarps) and pink (bodies) polygons (reference map). This is particularly evident when considering the accuracy of landslide size and FPr. These results are assessed more thoroughly in Figure 6 and Section 4.3.

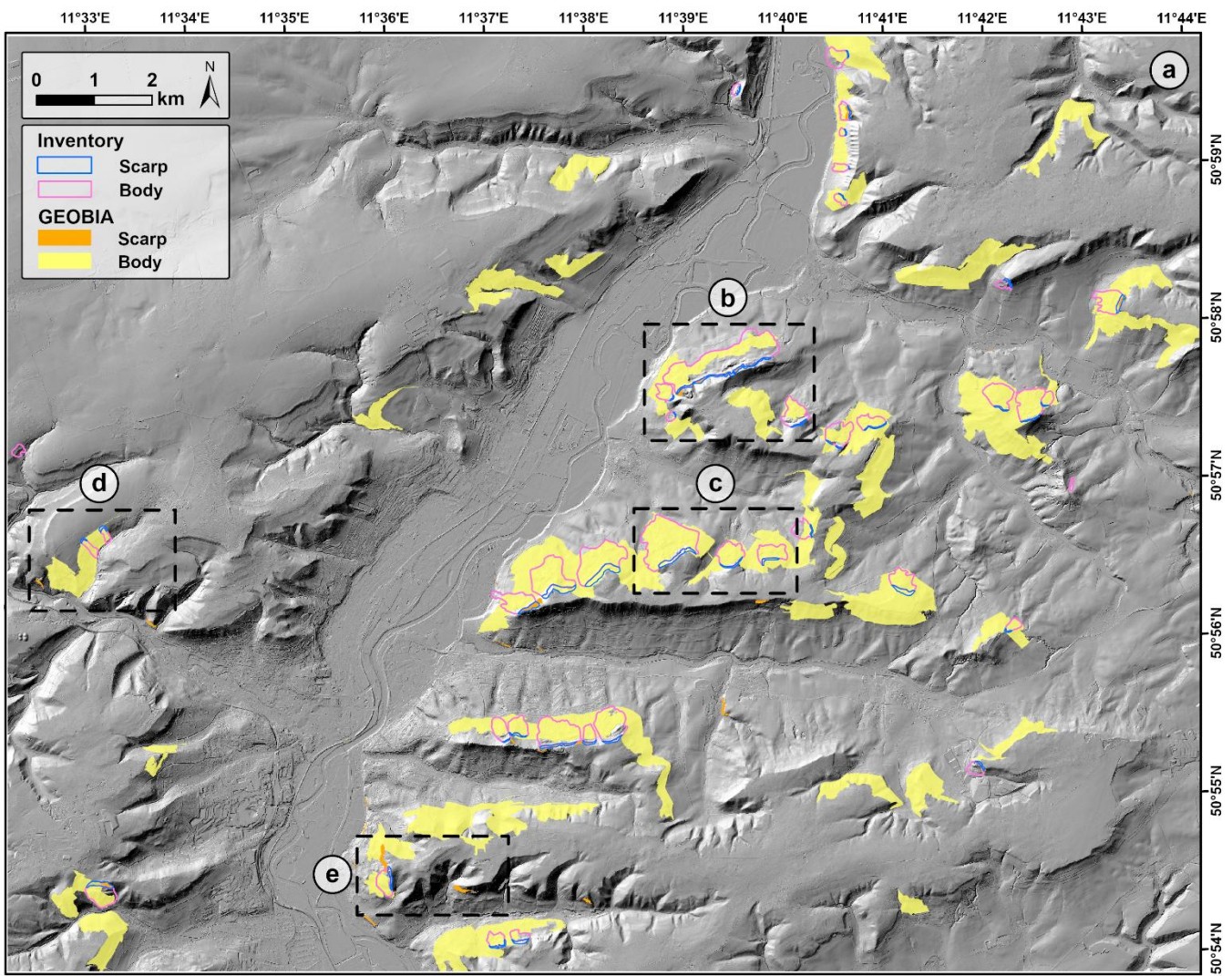

**Figure 4.** Final map of semi-automatic landslide detection using MI, displayed over hillshade DTM (TLUBN, 2019) throughout study area. Detected landslides are shown as coloured polygons (brown: scarps; yellow: bodies), while the inventory map is also displayed for comparison. The insets outlined by black dotted lines are magnified and analysed in Figure 8.

The morphological characteristics of the landslides (scarps and bodies) in the Jena region highlight notable distinctions between these components, as summarized in Tables 2 and 3. The results show that scarps have slope mean values of 48° and 40.5° in MI and MII, respectively, whereas bodies maintain a consistent slope mean value of approximately 20° across both models. This suggests that scarp areas are more sensitive to the optimized approach applied in MII, while body

areas remain relatively unchanged. Scarps exhibit positive values for plan curvature (C_plan), while bodies show lower values. Similarly, for profile curvature (C_profile), scarps display positive values, whereas bodies exhibit negative values, further emphasizing their distinct morphological characteristics. The TRI mean values further differentiate scarps and bodies, with scarps showing significantly higher values (3.73 and 5.09 for MI and MII, respectively) compared to bodies (0.95 and 1.93 for

270     MI and MII, respectively). Additionally, TPI values for scarp areas are higher in MII than in MI, while for body areas, the TPI values are notably lower in MII compared to MI.

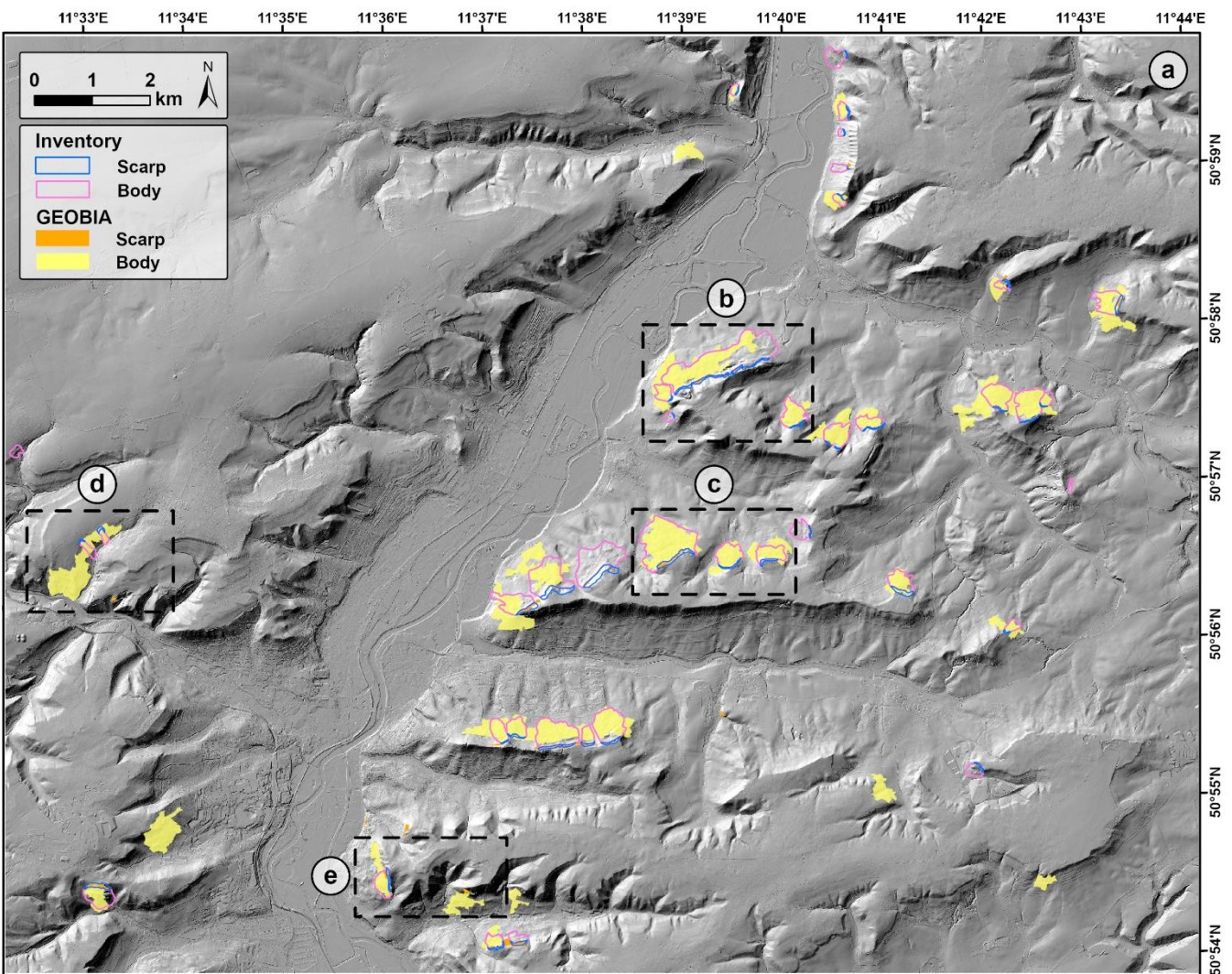

**Figure 5.** Final map of semi-automatic landslide detection using MII, displayed over hillshade DTM (TLUBN, 2019) throughout study area. Detected landslides are shown as coloured polygons (brown: scarps; yellow: bodies), while the inventory map is also displayed for

275     comparison. The insets outlined by black dotted lines are magnified and analysed in Figure 8.

Table 2. The morphological characteristics of landslides (scarps and bodies) in Jena region as detected by GEOBIA, Model I.

| LSVs/value | Scarp | | | Body | | |
|---|---|---|---|---|---|---|
| | min | max | mean | min | max | mean |
| Slope (°) | 34.9 | 61.5 | 48.2 | 14.88 | 25.69 | 20.28 |
| C_plan | -1.04 | 12.19 | 5.57 | -0.032 | 0.128 | 0.048 |
| TO3x3 | -0.003 | 0.029 | 0.013 | -0.00085 | 0.00051 | -0.00017 |
| TO25x25 | -0.001 | 0.099 | 0.049 | -0.00201 | 0.00373 | 0.00086 |
| TRI | 1.82 | 5.64 | 3.73 | 0.68 | 1.22 | 0.95 |
| TPI | 0.1 | 2.8 | 1.4 | -0.07 | 0.05 | -0.01 |

Table 3. The morphological characteristics of landslides (scarps and bodies) in Jena region as detected by GEOBIA, Model II

| LSVs/value | Scarp | | | Body | | |
|---|---|---|---|---|---|---|
| | min | max | mean | min | max | mean |
| Slope (°) | 33.4 | 47.7 | 40.5 | 15.9 | 27.9 | 21.9 |
| C_plan | 0.0001 | 0.0205 | 0.0107 | -0.0044 | 0.0014 | -0.0015 |
| C_profile | -0.0028 | 0.0040 | 0.0006 | -0.0014 | -0.0001 | -0.0012 |
| TO | -0.0014 | 0.0086 | 0.0036 | -0.00186 | 0.00003 | -0.00091 |
| TRI | 3.77 | 6.42 | 5.09 | 1.32 | 2.54 | 1.93 |
| TPI | 1.59 | 13.24 | 7.41 | -3.83 | 0.11 | -1.86 |

## 4.3 Accuracy assessment results

### 4.3.1 Thematic accuracy assessment

The thematic accuracy assessment is conducted using two complementary approaches: object-based (number-based) and area-based accuracy metrics. First, the number-based AA results (Table 4) show a significant improvement in the scarp zones for MII, with a higher TPr and significant reductions in both FPs and FNs. In the landslide body areas, the FPs were also significantly reduced, indicating an overall improvement in classification accuracy under the optimised window size.

Next, the area-based AA results are summarised in Table 5, with MII showing significant improvements over MI. As for scarp detection, MII increases the TP area from 6.3 ha to 9.1 ha and reduces the FNs accordingly, demonstrating better detection performance than MI. Although the FPs remain relatively high, they moderately decrease under MII. As for landslide body detection, MII significantly reduces the FPs from over 760 ha in MI to approximately 155 ha, indicating a significant improvement in mapping accuracy. However, a trade-off is observed, with the FNs increasing slightly. These trends are also noticeable in Figures 6 and 7, which shows a significantly lower FP in MII (red polygons in Figure 7) than in MI (Figure 6).

**Table 4.** Number-based comparison of True Positives (TPs), False Positives (FPs) and False Negatives (FNs) in landslides detection per model.

| Landslide | Model | TP | (%) | FP | (%) | FN | (%) |
|---|---|---|---|---|---|---|---|
| Scarps | I | 21 | 55.2 | 24 | 53.3 | 17 | 44.8 |
| | II | 25 | 65.8 | 5 | 16.6 | 13 | 34.2 |
| Bodies | I | 17 | 42.5 | 21 | 52.5 | 23 | 57.5 |
| | II | 21 | 52.5 | 8 | 20.0 | 19 | 47.5 |
| Inventory | 38 (scarps) / 40 (bodies) | | | | | | |

**Table 5.** Comparison of True Positives (TPs), False Positives (FPs) and False Negatives (FNs) in landslides detection per model (area in hectares).

| Landslide | Model | TP | % | FP | % | FN | % |
|---|---|---|---|---|---|---|---|
| Scarps | I | 191.2 | 78.7 | 769.2 | 80.0 | 51.8 | 21.3 |
| | II | 161.9 | 66.6 | 155.8 | 49.0 | 81.0 | 33.3 |
| Bodies | I | 6.3 | 30.6 | 9.3 | 59.6 | 14.3 | 69.3 |
| | II | 9.1 | 43.8 | 8.0 | 47.0 | 11.6 | 56.2 |
| Inventory | 20.7 (scarps)/ 243 (bodies) | | | | | | |

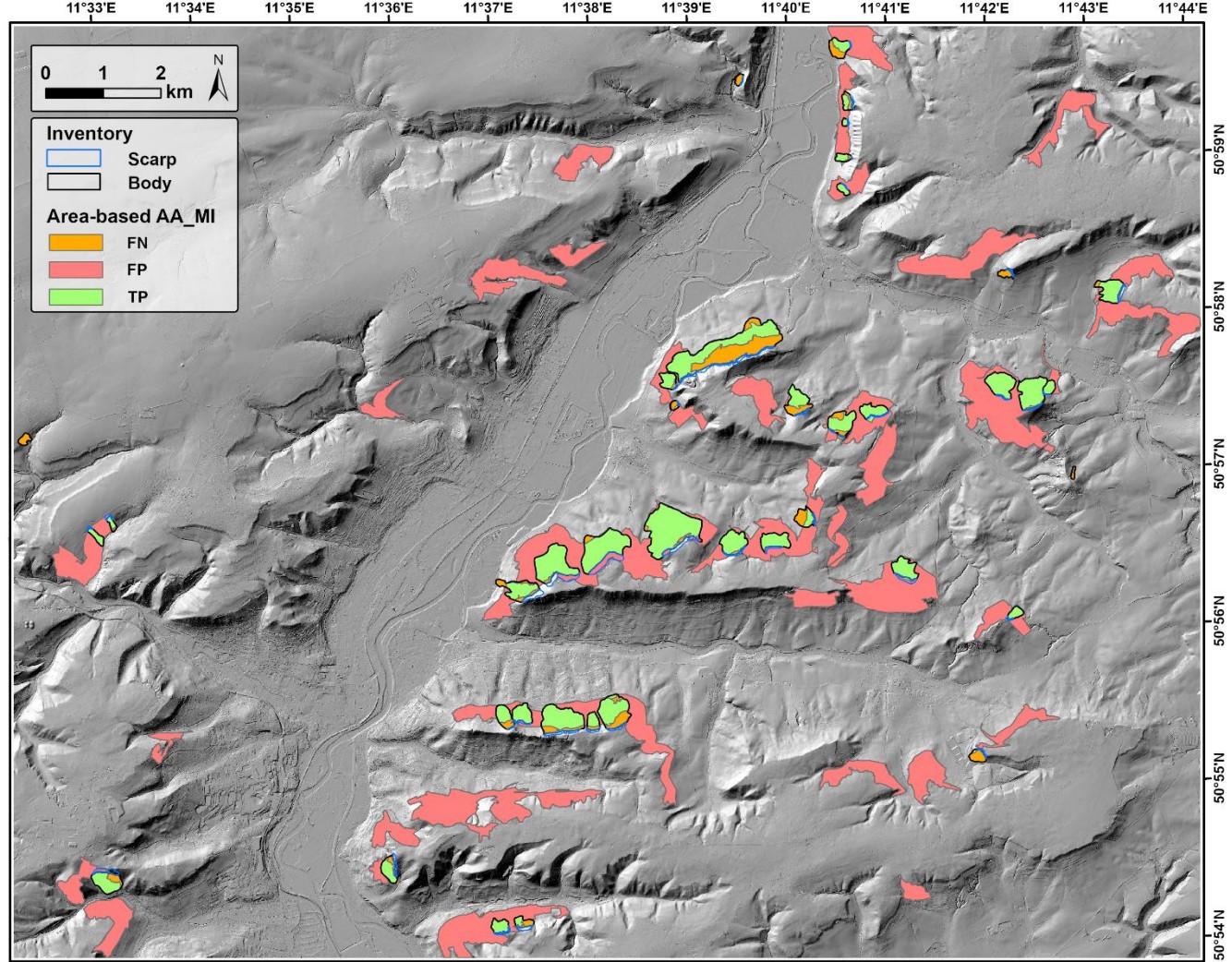

**Figure 6.** Area-based accuracy assessment of landslide body mapping using GEOBIA in Model I (True Positives (TP), False Positives (FP), and False Negatives (FN)), displayed over hillshade DTM (TLUBN, 2019).

### 4.3.2 Segmentation metrics

The US values for both models are below 0.50 in landslide scarp detection. Specifically, the US value drops from 0.84 in MI to 0.60 in MII, demonstrating that the use of the optimal window size in MII significantly improves the results (Table 6). Likewise, the OS value for scarps decreases from 0.53 in MI to 0.33 in MII. Additionally, the precision value improves in MII, confirming the effectiveness of the optimised approach.

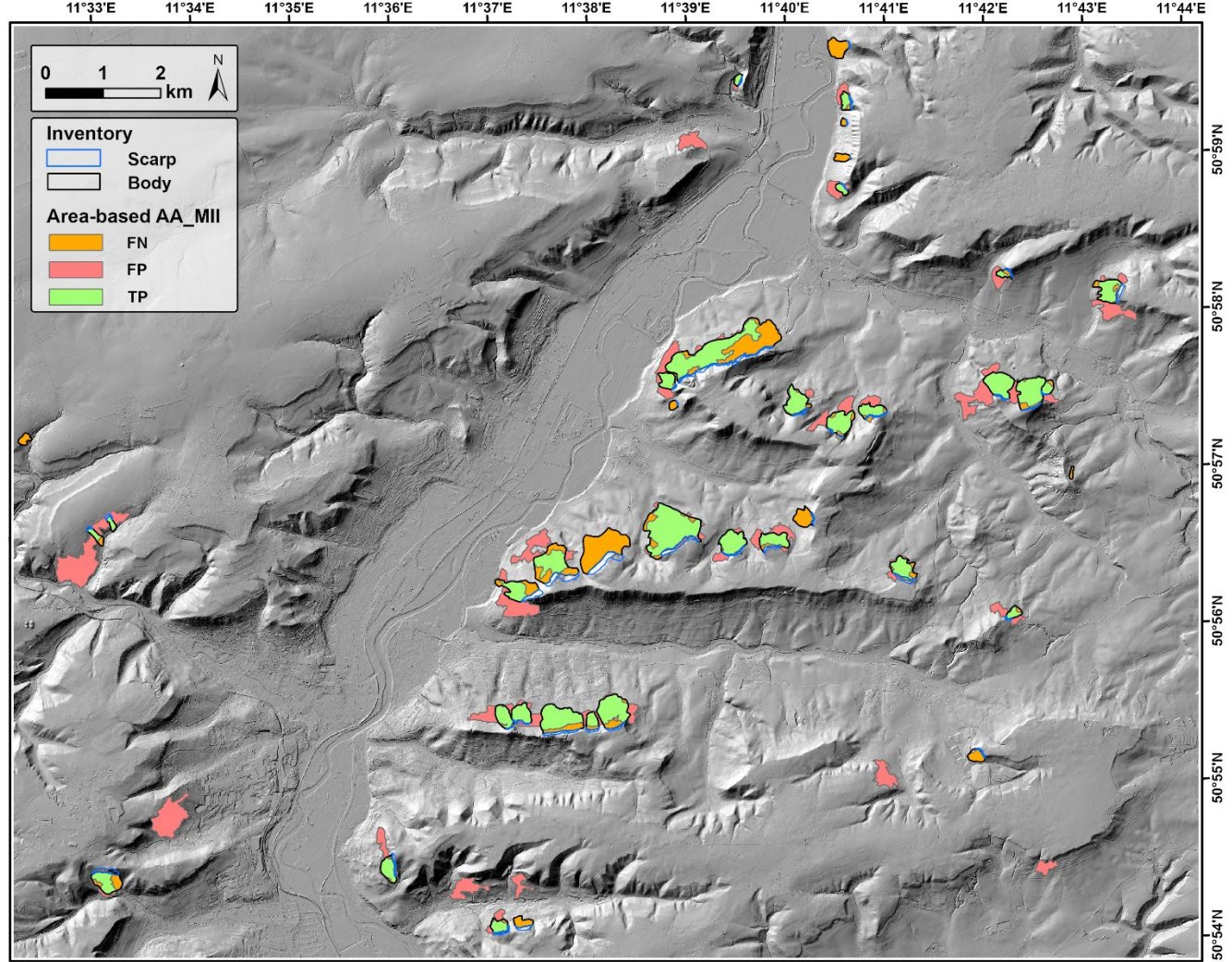

**Figure 7.** Area-based accuracy assessment of landslide body mapping using GEOBIA in Model II (True Positives (TP), False Positives
(FP), and False Negatives (FN)), displayed over hillshade DTM (TLUBN, 2019).

## 5 Discussion

This study highlights the efficacy of integrating GEOBIA with high-resolution DTM data for the inventory mapping of forest-
covered old landslides in middle-mountain regions. The implementation of the optimal window size (MII) substantially
enhances landslide detection accuracy while significantly reducing both the number and total area of FPs relative to the use of
the default window size (MI). These results align with those of Sîrbu et al. (2019), who demonstrated window size variability
across different LSVs for landslide scarps. Our study broadens this understanding by analysing both scarp and landslide body
areas, revealing that window sizes differ not only across LSVs but also between landslide scarp and body areas within the

same LSVs (Figure 3, Table 1). Consequently, landslide scarps and bodies should be detected separately within the model for an accurate analysis of each landslide component.

In our study area, large forest-covered landslides (>0.5 ha) are more successfully detected than smaller landslides, as they mostly show a strong geomorphological signature (Figure 8(b)). This is similar to previous findings (Knevels et al., 2019; Dias et al., 2023). However, MII has a lower proportion of misclassified areas within the same inset (Section 4.3.1, Figure 7), highlighting its ability to delineate landslide features accurately throughout the study area. Additionally, some automatically delineated landslide polygons extend beyond the boundaries of the inventoried actual landslide areas (Figures 4 and 5). These

deviations can be attributed to limitations in the segmentation and classification algorithms, which may produce irregular or overly coarse objects, or potential misclassifications or errors in object merging due to the use of a low threshold for object characteristics during refinement and the final layout.

    We should acknowledge that in one instance (see Figure 8(b)), a large mapped landslide may in fact consist of multiple individual events. Due to uncertainty regarding the precise boundaries between these possible landslides, we decided to map

the area as a single, large landslide in the inventory map. However, this issue is partially addressed through the area-based accuracy assessment presented in Section 4.3.1, which minimizes the impact of such limitations on model validation. Future studies could improve the accuracy further by labelling such ambiguous cases as "uncertain" or separating them from clearly defined landslides in the inventory. This would help to better assess model performance and transferability.

    Other misclassifications occur in areas where the geomorphological signature or roughness resembles the

characteristics of landslide body or scarp candidates defined by the developed rule set. Both models incorrectly detect a scarp in the same inset/window in Figure 8(d) but in different locations. The parameters in these areas are similar to those of actual landslide scarps, so distinguishing them solely based on DTM data is difficult. The misclassified area is actually rock outcrops, which are common in this region due to its local-scale variations in lithology, where different layers and materials respond differently to weathering and erosion (see the geological map of the region in Zangana et al., 2023). Most landslide scarp FPs

in our study can be attributed to this issue. In addition, both models misclassify landslide scarps as landslide bodies for two landslides from the inventory map in the same location in Figure 8(d). Thus, the scarp and landslide body areas in these cases landslides from the inventory map in the same location in Figure 8(d). Thus, the scarp and landslide body areas in these cases are highly similar, possibly due to landslide type/age and human activity, so the rule set cannot easily differentiate them from the rest of the study area. However, the landslides are successfully mapped, highlighting the need for detecting different

landslides separately. This shortcoming of our approach is similar to those reported in previous studies on forest-covered landslides (Li et al., 2015; Pawłuszek et al., 2019), demonstrating the challenges of performing inventory mapping without FPs using DTM data alone (Eeckhaut et al., 2012; Bell et al., 2012; Goetz et al., 2014; Knevels et al., 2019).

    Furthermore, both models identify the same location in Figure 8(e) as a landslide candidate (as a landslide scarp or a landslide body). The right-hand part of Figure 8(e) in MII shows a landslide that is excluded from the initial inventory due to

its dissimilarity with the other landslides and its small size. In MII, the body area of this landslide is incorrectly mapped over cropland, grassland and built-up areas. On the left-hand side of the same window (Figure 8(e), MI), the landslide body is wider

**Table 6.** Metrics used per landslide component per model.

| Metric | Scarp | | Body | |
|---|---|---|---|---|
| | M I | M II | M I | M II |
| AFI | -0.37 | -1.25 | -10.21 | -3.89 |
| OS | 0.53 | 0.33 | 0.18 | 0.18 |
| F_measure | 0.42 | 0.50 | 0.28 | 0.57 |
| US | 0.30 | 0.43 | 0.84 | 0.60 |
| recall | 0.36 | 0.48 | 0.80 | 0.77 |
| precision | 0.50 | 0.52 | 0.17 | 0.45 |

than the inventory map area that may correspond to the inventory map but is excluded from the initial inventory due to the insufficiency of the available information for it to be classified as a large landslide. In the MI image in Figure 8(d), the FP area extends over a wide region, particularly into forested areas and erosional rims, demonstrating that this model has lower landslide detection precision than MII. Therefore, landslide size should be addressed in future studies. Detecting landslides of different types and sizes at various levels can enhance detection rates and further reduce the FPr.

An analysis of the AA results reveals that MII outperforms MI in detecting landslide components (bodies and scarps) in terms of correctly identified areas and FP reduction. For instance, MII reduces the FPr by approximately 30% and 20% for landslide scarps and landslide bodies, respectively (Table 4). MII slightly underestimates the landslide body areas; the TPr for landslide body detection decreases, offset by an increase in the TPr for scarp areas. Although MI yields a higher TPr for landslide body areas, its FPr is approximately 30% higher than that of MII, highlighting the importance of considering the FPr alongside the TPr and FNr for a comprehensive evaluation of model performance. MII has considerable potential, effectively reducing the FPr while maintaining a strong TPr. However, further research is needed to enhance the applicability of this approach across various environments and datasets.

Our TPr is comparable to that of Eeckhaut et al. (2012), who investigated the semi-automatic detection of landslides using lidar data and identified 71% of landslides using object-based detection. In comparison, MII achieves TPr values of 65.8% and 72.5% for landslide scarps and landslide bodies, respectively. However, our approach demonstrates significantly better accuracy through minimising of FPs. Eeckhaut et al. (2012) identified 18 FP landslides in an area affected by 38 deep-seated landslides (47%), and Martha et al. (2010) reported 73 FP landslides for 55 expert-identified landslides (132%). As shown in Tables 2 and 3, our study identifies only 5 FP landslide scarps (20% relative to the TPs) and 7 FP landslide bodies (24% relative to the TPs). Therefore, MII detects landslides more efficiently than MI and the abovementioned approaches. However, some studies (Martha et al., 2010; Eeckhaut et al., 2012; Knevels et al., 2019) do not provide the number or total

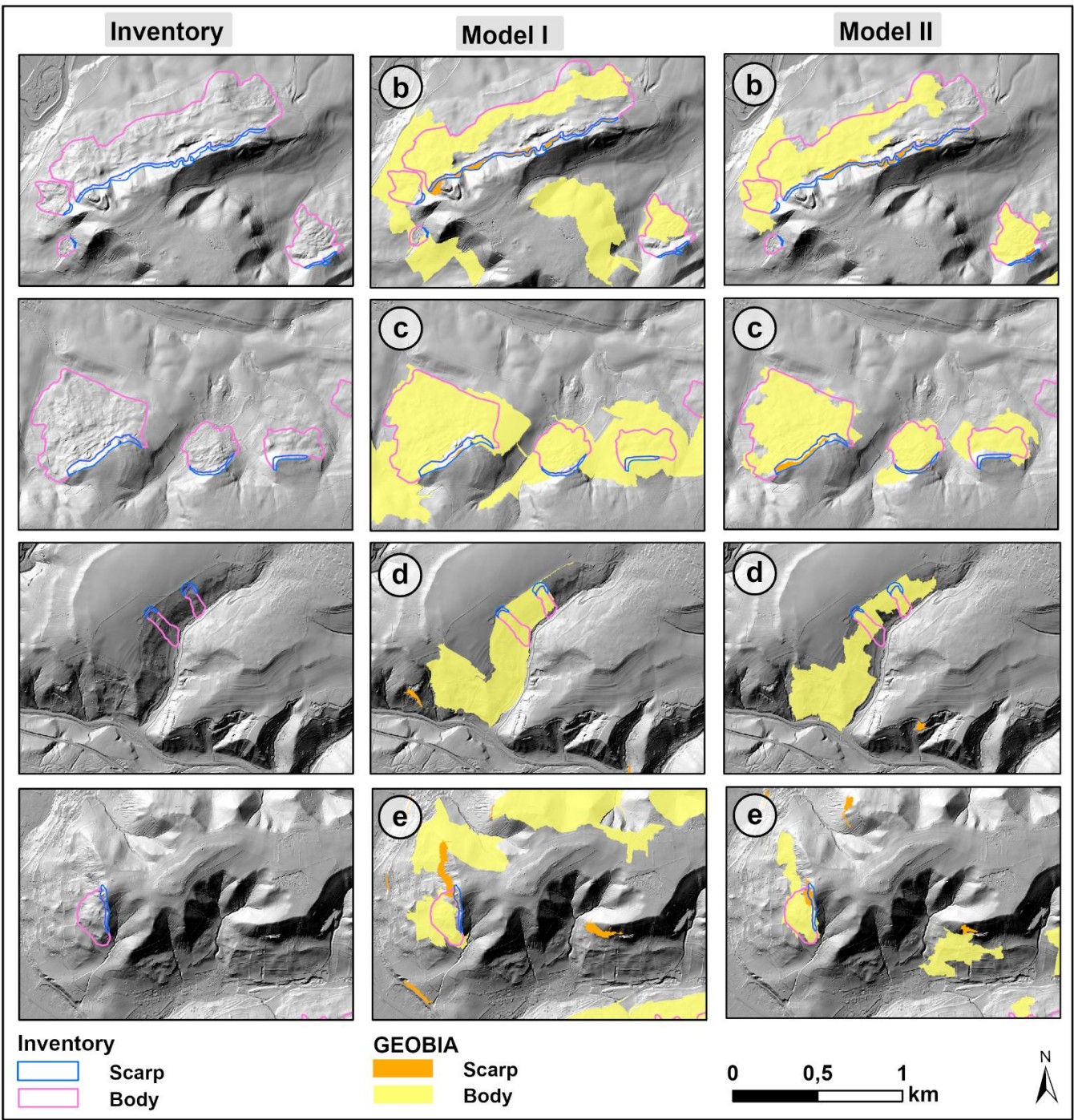

**Figure 8.** Detailed results of Model I (MI) and Model II (MII) for selected areas. The left column shows the landslide inventory overlaid on the hillshade DTM (TLUBN, 2019). The middle and right columns display MI and MII results, respectively, corresponding to the inset areas b–e shown in Figures. 4 and 5. These zoomed-in views highlight differences in model performance within representative subregions.

area of FPs or only present them numerically, hindering a comprehensive comparison of methods. Conversely, our evaluation is more detailed, enabling a systematic assessment of model precision in identifying a specific number of TPs while minimising FPs.

Misclassification also occurs in areas with degraded geomorphological signatures (Figure 8(d, e)). Landslide scarps are often misidentified in regions with features resembling those of scarp candidates. These errors may be attributed to human-made structures, such as roads, pits, ridges and rock toes. Similar false detections are observed in areas with partially eroded or weathered limestone formations, particularly in areas where they overlay sandstone. This misclassification pattern aligns with commonly reported results (Eeckhaut et al., 2012; Hölbling et al., 2017; Knevels et al., 2019; Dias et al., 2023).

Selecting the appropriate window sizes for the LSVs is challenging (Figure 3, Table 1). The results show considerable variability across the 10 runs, with the optimal window sizes often being larger for scarp zones than for body areas. For example, the TPI and TRI significantly deviate between the optimal and default window sizes, highlighting the need to fine-tune window sizes based on the specific geomorphometric characteristics of each landslide feature. By contrast, the default window sizes used in MI often result in under- or over-segmentation, especially in scarp zones, compromising detection accuracy. These results highlight the complexity of adjusting window sizes for accurate scarp and body detection and emphasise the advantages of the tailored approach of MII in overcoming these limitations.

Compared with MI, MII is more effective in detecting individual landslides as distinct polygons and separating them from neighbouring landslides. Additionally, MII significantly reduces the FP number and area, as shown in the model comparison in Figure 7. This highlights the robustness of the optimal window size approach of MII, which is more effective for landslide detection than default or manually selected window sizes, which have been commonly used in many studies for decades. Overall, this approach is a valuable advancement in improving semi-automatic landslide inventory mapping, particularly in forested areas, where the limited availability of optical data often hampers complete inventory mapping (Eeckhaut et al., 2012; Li et al., 2015; Knevels et al., 2019). Although Stage III was first developed in the Jena region, the framework and rulesets can be applied to other areas where landslides are well expressed in hillshade and distinct from their surroundings. This condition is often met for forest-covered landslides of varying ages, and may also apply to younger landslides, with no or limited human impact. While some refinement parameters may need local adjustment, the method is not restricted to a specific landslide type or setting, and its broader applicability should be further evaluated in future studies

## 6 Conclusion

A GEOBIA-based approach is developed and used for the semi-automatic mapping of landslide inventories in the forested areas of the middle-mountain regions of Jena, Germany. The proposed method effectively maps forest-covered landslides, with a particular focus on medium and large landslides (greater than 0.5 ha), but does not detect smaller landslides. MII, which uses the optimal window size, maps landslide scarps with higher accuracy than MI, which relies on default window sizes. MII shows a significant (15%) improvement in scarp detection during number-based AA while reducing the FPr by 30%. However,

this FPr reduction entails a trade-off, as the FNr increases by approximately 15%. Nonetheless, MII remains highly effective for semi-automatic landslide mapping in forest-covered areas.

High-resolution DTM derivatives serve as the base data for landslide mapping using GEOBIA, which incorporates an optimal window size to detect forest-covered old landslides in middle-mountain regions. Our analysis shows that this approach may significantly improve the accuracy of landslide mapping in areas with sparse or no vegetation and in regions where landslides are newly formed or have recently altered the terrain. This is due to the superior ability of DTM data to show recent landslide features compared with historical ones.

This study emphasises the importance of calculating window sizes separately for different landslide components, as landslide scarps and bodies require distinct window sizes for accurate detection. This factor should be considered thoroughly before any calculation or modelling process involving landforms of interest. As landform detection depends on the defined window size, our automated objective approach is highly suited for future research and semi-automatic landform modelling. However, further evaluation is required to ascertain the transferability of this method to other regions; nonetheless, this method should be globally applicable to the detection of landslides with well-defined geomorphological features using high-resolution DTM data. As the pioneering use of GEOBIA for landslide inventory mapping in the Jena area, this study serves as a foundation for future research on landslides in this region. Furthermore, it can be used as a base map for hazard and risk assessments, especially as climate change may reactivate old landslides, as has occurred in many areas across Germany and around the world.

Appendices

 **A) Ruleset Model I**

**Table A1: Landslide Scarps Refinement Steps**

| Refinement Step | Actions | Conditions / Criteria | Additional Notes |
|---|---|---|---|
| *Refinement 1* | Remove objects | IF SLOPE < 33 AND TRI < 2 | Then → Merge objects |
| *Refinement 2* | Expand objects | Remove IF TRI < 1.7<br>IF non-Scarp (Rel. Border to Scarp Candidate > 0.1 AND TRI > 2.5 AND SLOPE > 45)<br>Remove IF AREA < 1500 pixels AND Length/Width > 5 | Then → Merge objects |
| | Remove objects | IF TPI < -0.01 AND (SLOPE < 33 AND TRI < 2)<br><br>Remove IF (TRI < 1 AND TPI < 1)<br>Remove IF Length/Width > 3 AND TRI < 1.8<br>Remove IF Rectangular fit < 0.1 AND Roundness > 2 | Then → Merge objects |
| *Refinement 3* | Remove objects | IF (AREA > 20,000 pixels AND TPI < 0.1) OR AREA < 100 pixels | AND IF TPI < 0.02 OR Length/Width > 10 OR C_plan < 0 |
| | Remove objects | IF SLOPE < 34 OR C_plan < -3<br><br>Remove IF TOP > 0.03 OR TOP < -0.004<br>Remove IF TPI < 0.1 | Then → Merge objects |

**Table A2: Landslide Bodies Refinement Steps**

| Refinement Step | Actions | Conditions / Criteria | Additional Notes |
|---|---|---|---|
| *Refinement 1* | Expand objects | IF unclassIFied (SLOPE > 12 AND Rel. Border to Body Candidate > 0.5)<br>IF non-Body (Rel. Border to Body Candidate > 0.5 AND Rel. Border to non-Body ≤ 0.2)<br>IF non-Body (Rel. Border to Body Candidate > 0.6) | Then → Merge objects |

| | | | |
|---|---|---|---|
| | | IF non-Body (SLOPE > 10, C_plan < 0.1 AND Rel. Border to Body Candidate ≥ 0.1) | |
| | | Remove IF AREA < 5000 pixels | |
| | Expand objects | IF non-Body (Rel. Border to Body Candidate ≥ 0.1 AND C_plan < 0.2 AND SLOPE > 12) | |
| | | IF non-Body (Rel. Border to Body Candidate ≥ 0.4, C_plan < 0.3 AND SLOPE > 15) | |
| | | IF non-Body (Rel. Border to Body Candidate ≥ 0.7 AND C_plan < 0.3 AND SLOPE > 10) | |
| | Remove objects | IF Length/Width ≥ 6.1 OR (Length/Width ≥ 5 AND SLOPE < 15) | Then → Merge objects |
| | | Remove IF AREA < 15,000 pixels | |
| *Refinement 2* | Remove objects | IF SLOPE < 14 AND TRI < 0.6 | |
| | Expand objects | IF unclassIFied (Rel. Border to Body Candidate ≥ 0.5 AND SLOPE ≥ 12) | |
| | | IF non-Body (SLOPE > 10, C_plan < 0.1 AND Rel. Border to Body Candidate ≥ 0.1) | |
| | | Remove IF non-Body (Rel. Border to Body Candidate > 0.5 AND Rel. Border to non-Body ≤ 0.2) | Then → Merge objects |
| | | Remove IF (TRI < 0.7 AND TPI < -0.05) OR AREA < 15,000 pixels | |
| | Expand objects | IF unclassIFied (Rel. Border to Body Candidate = 1) | Then → Merge objects |
| | | → Expand IF (TRI > 0.6 AND C_plan > 0.3) | |
| | | Remove IF AREA < 46,000 pixels | |
| | Expand objects | IF unclassIFied (Rel. Border to Body Candidate > 0.5, C_plan > 0.1 AND TRI > 0.45) | OR IF Rel. Border to Body Candidate > 0.5 → Merge objects |
| | Remove objects | IF (AREA < 150,000 pixels AND C_plan < 0) OR SLOPE < 5 | |
| *Refinement 3* | Remove objects | IF SLOPE < 14 | |
| | | IF TPI < -0.1 | |
| | | IF C_plan < -0.04 OR C_plan > 0.2 | Then → Merge objects |


## B) Ruleset Model II

**Table B1: Model II Landslide Scarps Refinement Steps**

| Step | Action | Conditions / Criteria | Notes |
|---|---|---|---|
| *Initial Pre-processing* | Segmentation | MRS 20 (Shape: 0.1, Compactness: 0.5) | |
| | Initial classIFication | Remove plateau AND floodplain AREAs by identIFying "Initial Scarp AREA" to retain Landslide polygons AND buffers | |
| *Potential Scarp IdentIFication* | Remove segments | IF mean SLOPE ≥ 0.3, TPI ≥ 0.65, TRI ≥ 0.8 | Define as Potential Scarp |
| | Expand AREA | IF mean SLOPE > 0.7 AND Rel. Border to Potential Scarp ≥ 0.45 Or mean SLOPE ≥ 25, TPI ≥ 1.5, mean TRI ≥ 2.5 | |
| | ClassIFication | SVM (Scarp vs Non-Scarp) | |
| *Refinement 1* | Expand Scarps | IF Rel. Border to Scarp > 0.85 AND mean SLOPE ≥ 0.4 Or Rel. Border to Scarp > 0.3, mean TPI ≥ 12, mean SLOPE > 0.6 | |
| | Remove Scarps | IF −29 < mean C_profile > 6.3 AND −5.2 < mean C_plan > 37 IF −0.04 < TOP > 0.14 IF mean SLOPE > 0.45 AND mean TPI < 0.14 or mean TRI < 1.14 | |
| | Further Expand Scarps | IF mean SLOPE > 0.65, mean TRI > 2.2, mean TPI > 7, AND Rel. Border to Scarps > 0.2 For unclassIFied/non-Scarp: mean SLOPE > 0.4 AND Rel. Border to Scarps > 0.9 | |
| | Remove Scarps | IF Rel. Border to Scarps < 0.4 AND AREA > 2000 pixels | |
| | Merge segments | Merge all segments | |
| *Refinement 2* | Remove Scarps | IF AREA < 1000 pixels AND mean SLOPE < 0.74 | |
| | Expand Scarps | IF mean SLOPE > 0.45, Rel. Border to Scarps > 0.25, Rel. Border to non-Scarps < 0.01 Or mean SLOPE > 0.7, Rel. Border to Scarps > 0.3, mean TRI < 2.15 Or mean SLOPE > 0.65, Rel. Border to Scarps > 0.1, mean TRI < 2.1, Rel. Border to non-Scarps < 0.5, AREA < 6800 pixels | |
| | Merge non-Scarps | IF mean SLOPE > 0.48, TRI > 1, mean TPI > 7 | |
| | Further Expand Scarps | IF mean SLOPE > 0.5, Rel. Border to Scarps > 0.3, Rel. Border to non-Scarps > 0.43, TRI < 1.7 | |
| | Remove Scarps | IF Rel. Border to non-Scarps > 0.85, AREA < 3000, mean SLOPE < 0.7 AND IF mean TPI < 2.45, mean SLOPE < 0.65, AREA < 1600 Or mean SLOPE < 0.75, AREA < 8000, mean TPI > 10 Or mean SLOPE < 0.7, AREA < 10,000, mean TOP < 0.028 | |
| *Landslide Bodies* | | | |

 **Table B2: Model II Landslide Bodies Refinement Steps**

| Step | Action | Conditions / Criteria | Notes |
|---|---|---|---|

| | | |
|---|---|---|
| *Initial Pre-processing* | Segmentation | MRS 30 (Shape: 0.1, Compactness: 0.5) |
| | Potential Body Segments | Remove IF mean SLOPE ≥ 0.2, TPI < 7, TRI ≥ 0.5 |
| | ClassIFication | SVM (Bodies vs Non-Bodies) |
| *Refinement 1* | Expand Bodies | IF Rel. Border to Body ≥ 0.8 AND mean SLOPE > 0.25, or Rel. Border ≥ 0.9 |
| | Merge Body segments | Merge all Body segments |
| | Expand further | IF Rel. Border to Body ≥ 0.7, Rel. Border to non-Body < 0.3, mean SLOPE ≥ 0.25 |
| | | Or Rel. Border to Body ≥ 0.8, Rel. Border to non-Body < 0.3, mean TRI ≥ 0.35 |
| | | Or Rel. Border to Body ≥ 0.4, Rel. Border to unclassIFied < 0.01 |
| | | Or Rel. Border to Body ≥ 0.55, Rel. Border to non-body < 0.2, mean TRI > 0.6 |
| | | Or Rel. Border to Body ≥ 0.6 AND mean TRI > 0.4 |
| | | Or Rel. Border to Body ≥ 0.15, mean TRI > 1, mean SLOPE > 0.4 |
| *Refinement 2* | Merge Body segments | Merge all Body segments |
| | Expand Bodies | IF Rel. Border to Body ≥ 0.3, mean SLOPE > 0.2, mean TRI > 0.6 |
| | Remove Bodies | IF −.0022 < mean TOP ≥ 0.003 |
| | | Or AREA < 10,000 pixels AND mean SLOPE > 0.25 |
| | | Or AREA < 25,000 pixels AND mean SLOPE < 0.25 |
| | | Or AREA > 400,000 pixels |
| | | Or mean C_plan < 0.0 AND/or mean C_profile < 0.0 AND AREA > 50,000 or mean TPI > 1.3 |
| | Expand from unclassIFied/non-Body | IF Rel. Border to Body > 0.99 |
| | Remove Body segments | IF Rel. Border to non-Body > 0.99, or Rel. Border > 0.9 AND mean SLOPE < 0.29 |
| *Refinement 3* | Remove Bodies | IF mean SLOPE < 0.4 AND AREA < 15,000 pixels |
| | | Or mean SLOPE < 0.3, AREA < 50,000, mean TPI > 0.8 |
| | | Or mean SLOPE < 0.42 AND mean TPI < -1.9 |
| | | Or AREA < 40,000 AND mean TPI > 1.4 |
| | | Or mean C_profile < 0.0 AND mean TPI > 1, or mean SLOPE < 0.25 |
| | | Or 15,000 < AREA < 65,000 AND Length/Width > 1.99 |
| | | Or mean C_profile > -0.05 |
| | | Or 15,000 < AREA < 100,000 AND 3.2 > Length/Width > 1.99 |
| | | Or 25,000 < AREA < 100,000, or C_plan < -0.02, or C_plan > 0.19 AND mean TPI < 1 |
| | | Or St. Deviation of C_profile < 1.6 |
| | Expand Body AREA | From unclassIFied segments IF Rel. Border to Body > 0.6, mean SLOPE > 0.4, mean TRI > 1 |
| | | Or Rel. Border to Body > 0.42 AND Rel. Border to unclassIFied = 0 |
| | | Or from non-body AREAs IF Rel. Border to Body > 0.7 |
| | Remove Bodies | IF AREA > 183,000 pixels AND mean TRI > 0.81 |

*Code availability.* The code used in this research is available from the first author upon reasonable request. The first version related to optimal moving-window sizes was previously published by Sîrbu et al. (2019).


*Data availability.* The resulting data used in this research can be obtained from the first author upon reasonable request but can be accessed through the references in the methodology section.

*Author contributions.* IZ: conceptualisation, data curation, formal analysis, visualisation, writing (original draft, review and editing). RB: conceptualisation, visualisation, writing (review and editing). LD: conceptualisation, data and software implementation, writing

(review and editing). FS: developed and calibrated the original R script to obtain the optimal moving-window sizes, writing (review and editing). LS: conceptualisation, writing (review and editing), supervision. All authors have read and agreed to the published version of the paper.

*Competing interests.* The contact author has declared that none of the authors has any competing interests.

*Disclaimer.* Publisher's note: Copernicus Publications remains neutral with regard to jurisdictional claims made in the text, published maps, institutional affiliations, or any other geographical representation in this paper. While Copernicus Publications makes every effort to include appropriate place names, the final responsibility lies with the authors.

*Acknowledgements.* Preliminary results of this work have been presented to the EGU2024 Conference. We would like to thank Melanie

Stammler and Simon Terweh for their valuable discussions.

*Financial support.* This work was supported by the Open Access Publication Fund of the University of Bonn.

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
