# Peer review of "Inventory mapping of forest-covered landslides using Geographic Object-Based Image Analysis (GEOBIA), Jena region, Germany"

_EGUsphere, 2025_

## Author Comment (AC1)

**Inventory mapping of forest-covered landslides using Geographic Object-Based Image Analysis (GEOBIA), Jena region, Germany**

Authors' response

Anonymous Referee #1

Comment:

This study presents a semiautomatic method for landslide identification in Germany. I find the topic relevant and promising; however, improvements are necessary, particularly in the methodology section, which requires a more detailed description. Additional comments and suggestions are outlined below:

1. The authors use the term "landslides" in the introduction. In English, this is a general term encompassing all types of mass movements (e.g., shallow landslides, debris flows, rockfalls, etc.). Did your analysis identify all these types? If not, I recommend using a more precise term to reflect the specific process addressed in the study.

Response:

Thank you for the comment. In this study, we did not map all types of mass movements, but primarily focused on deep-seated (rotational) landslides, along with a few old shallow landslides. However, we did not differentiate between landslide types in our analysis, basically, the focus was on forest-covered landslides in general. Unfortunately, using LiDAR DTM, we do not have any information on their age. However, assuming that features are well preserved under forest cover, they might be quite old (Bell et al., 2012). We agree that clarification is helpful, and we have added a brief explanation regarding the mass movements we study in the last paragraph (revised manuscript, at line 83) of the Introduction.

2. **Study Area section**: Please provide information on recorded damage and economic losses in the region, if available. What is the primary triggering factor for landslides in Thuringia? Is it related to tectonic activity, climatic conditions, or other factors?

Response:

Thank you for this valuable comment. Unfortunately, specific information on recorded damage and economic losses in the region is not available, and we have included this information in the revised manuscript, at line 115-116.

To address the second concern, we have expanded the 'Study Area' section to clarify the main triggering factors. The updated text explains that landslides in this region are primarily caused by geological and structural conditions, particularly where limestone (Muschelkalk) overlies sandstone (Buntsandstein), and by steep slopes along the cuesta scarp. We also acknowledge that permafrost thawing at the end of the last glaciation may have played a significant role in triggering many of the older, deep-seated landslides (Achilles et al., 2016). These clarifications have now been incorporated into the revised manuscript, at lines 105-111.

3. **Line 98**: The manuscript states that "the area has experienced periods of landslide activity." Please specify which periods are being referred to.

**Response:**

Thank you for pointing this out. In our study, we focus on old, forest-covered landslides identified through DTM-based mapping and semi-automatic analysis. The study by Achilles et al. (2016) assumes that the landslides may have occurred during the Holocene, possibly beginning at the end of the last glaciation. Although the DEM, as our primary data source, does not provide explicit temporal information, the presence of large landslide features beneath dense forest cover suggests that there may have been a period of increased landslide activity in the past. While we are unable to determine the precise timing of these events, this interpretation aligns with the assumptions discussed in the above mentioned research. This information has been incorporated into the revised manuscript, at lines 109-114.

4. **Data section**: What criteria were used for visual landslide mapping? Which types of landslides were identified and mapped? This information is essential and should be included.

**Response:**

Thanks for this comment. In the revised manuscript, at lines 132-143, we have clarified the criteria used for visual landslide mapping, as well as the types of landslides identified. The landslide inventory map (reference map) was produced through manual visual mapping in ArcMap 10.7, primarily based on traditional and multi-directional hillshade derived from 1m LiDAR–DTM data. This method follows the procedure described by Schulz (2004), which is already cited.

Although hillshade was the only data type directly used to create the inventory, additional land-surface variables (LSVs), such as slope, curvature

(plan and profile), topographic openness, topographic position index (TPI) and terrain ruggedness index (TRI), were employed to assist with on-screen interpretation. These LSVs were particularly useful for improving the delineation of landslide boundaries where hillshade alone did not provide sufficient contrast. In most cases, the landslide scarp and body were mapped separately if they could be visually distinguished; however, in a few instances, identification of the scarp was not possible.

The inventory mainly consists of deep-seated (rotational) landslides, with some shallow features also present. However, the study does not explicitly classify landslide types, as the objective is to detect medium to large forest-covered historical landslides.

5. **GEOBIA-based landslide inventory mapping section**: Please specify the versions of the GIS software used (e.g., eCognition, ArcGIS).

**Response:**

Agree, and we have specified them in the revised manuscript, at line 145 and 148.

6. **Figure 1**: I suggest incorporating the symbol for landslide features (currently shown in white) into the **map legend** itself rather than only in the figure caption. This will enhance immediate understanding, as the current legend indicates landslides in green, which is confusing.

**Response:**

Thanks for pointing this out. We totally agree. Therefore, the map legend revised to insure consistency between the legend and the figure caption.

7. **Line 126 – Step 1**: What were the specific criteria applied for visual landslide mapping? This information is crucial. Please also state the total number of landslides identified and the total mapped area.

**Response:**

Thank you for pointing this out. The first part of your comment appears to overlap with point 4 of your earlier comments, which we have already addressed. The second part of your concern has been addressed in Section 4.3, which contains all the relevant information. Specifically, Table 4 in the revised manuscript shows the number of mapped landslides, note the last rows marked "**Inventory** = **38**" and "**40**" for scarps and bodies, respectively. Table 5 (in the revised manuscript, at the last row) presents the total mapped area (the inventory) in hectares: **20 ha** for scarps and **243 ha** for bodies. This information was used to evaluate and compare the results of our method with those from manual mapping. To avoid redundancy, we have not

repeated these details elsewhere in the manuscript. We hope this clarification adequately addresses your concerns.

8. **Line 136:** Indicate the software versions used for ArcGIS and R.

**Response:** Thank you, we agree and have incorporated it in the revised manuscript at line 153.

9. **Lines 145–150:** What were the proportions of the samples used for landslide scarps, landslide bodies, and non-landslide areas? Please include this breakdown.

**Response:**

Thanks for this comment. We agree with you and have made the necessary revisions. The proportions of each sample are included in the revised manuscript, at lines 158-159. It is also important to note that the non-landslide areas have been divided into two categories: non-scarp and non-body areas. This means that they are not treated as a single, non-landslide class.

10. **STAGE II – Segmentation and Classification:** Include the segmentation parameters such as **shape** and **compactness**.

**Response:** Done.

**Suggestion:** A figure illustrating the mapped landslide scarps and bodies would enhance clarity.

**Response:**

Thank you for your suggestion. Although the specific concern was not entirely clear to us at first, we decided to revise Figures 4, 5, and 7 (Fig 7 renumbered to Fig 8 in the updated version) to further improve clarity. In the updated figures, landslide scarps and bodies from the inventory and the GEOBIA results are represented using distinct colors, which we believe enhances visual interpretation and facilitates a clearer comparison. We hope these revisions address your point effectively.

11. **Line 162:** Please elaborate on the "refinement process." What specific criteria were used to determine when the result was satisfactory?

12. **Lines 162–166:** Provide the threshold values used in the ruleset applied during the classification (e.g. shape and compactness).

**Response:**

Thank you for these valuable comments on related issues (comments #11 and 12). As they both relate to the classification and refinement process, we have decided to respond to them together.

In the original manuscript (lines 162–164), we mention the general classification criteria used: '*In this phase, we utilised morphometric parameters of the LSVs and classified objects, including their mean values, standard deviations, length-to-width ratios, areas, relative borders, and distances to specific objects*.'

While this description outlines the general approach, we agree that a more specific and detailed explanation is necessary. Therefore, in the revised version of manuscript, we have expanded the explanation by include a comprehensive set of rule-set parameters in the **Appendices (Table A1-B2**). These clarify the exact criteria and thresholds applied during the classification process for both scarps and landslide bodies.

The iterative refinement process was guided by a combination of visual inspection and comparison against the manually mapped landslide inventory. At each step of the refinement process, we evaluated the outputs by analysing the number of true positives, false positives and false negatives, adjusting the threshold values accordingly in order to strike a balance between accuracy and minimising misclassifications, this already explained in other way in the Result and Discussion as well.

13. **Line 194**: (Dias et al., 2023). Ensure proper in-text citation formatting and consistency.

**Response:** Thanks for this comment. We agree and have checked through the manuscript.

14. **Figures**: Improve the resolution and overall size for better readability and visual interpretation.

**Response:**

Thanks for you for your comment. We agree that the figures should be clearly readable across all file types. Accordingly, we have modified the study area figure (Fig. 1), as already recommended, and we have also revised Figures 4, 5, and 7 (Fig 7 renumbered as Fig. 8 in the revised manuscript) to further improve clarity. In the updated figures, landslide scarps and bodies from the inventory and the GEOBIA results are depicted using distinct colors, which we believe improves visual interpretation and facilitates clearer comparison. Additionally, we have split the original Figure 6 into two separate figures (now Figures 6 and 7 in the revised manuscript) and revised them with higher resolution, larger dimensions, and improved layout. We have also updated the figure captions to be more detailed and descriptive. We hope these revisions effectively address your concerns (see the revised manuscript).

15. **Section 4.2 – GEOBIA-based landslide modeling results**: Include the total area identified for landslide features by both Method I and Method II for comparison.

**Response:**

We addressed this point in our response to comment 7, which we hope clarifies your concern. Please let us know if any further details are still required.

16. The **Results** section needs to be more comprehensive. Please include more descriptive analysis and interpretation of the findings.

**Response:**

Thank you for your comment. Following your recommendation, we have added two tables to **Section 4.2** ('GEOBIA-based landslide modelling results') to clarify the findings. The tables (Tables 2 and 3, revised manuscript) present morphological parameters for GEOBIA-based mapping results, including scarps and bodies, for both models. Brief interpretations of these results have also been integrated into the revised manuscript to enhance understanding, at lines 262 and 271.

17. There are two references listed for **Dias et al. (2023)**, labeled "a" and "b" in the references. However, only **Dias et al. (2023)** is cited in the main text. Please ensure that the correct designation (**a** or **b**) is used consistently in both the text and the reference list.

**Response:**

Thank you for your observation. For the record, I only intended to reference **the second article, or 'b'**, in the original version. Consequently, you will now see a single citation for **Dias et al. (2023)** in both the text and the reference list, **with no 'a'** or **'b'** designation.

**References:**

Achilles, F., Danigel, M., Frey, V. S., Voigt, T., and Büchel, G.: Massenbewegungen am Übergang des Oberen Buntsandstein in den Unteren Muschelkalk im Jenaer Gembdental, Beitr. Geol. Thüringen, NF.23, 103–114, 2016.

Bell, R., Petschko, H., Röhrs, M., and Dix, A.: Assessment of landslide age, landslide persistence and human impact using airborne laser scanning digital terrain models, Geogr. Ann. Ser. A Phys. Geogr., 94, 135–156, https://doi.org/10.1111/j.1468-0459.2012.00454.x, 2012.

Dias, H. C., Hölbling, D., and Grohmann, and C. H.: Rainfall-Induced Shallow Landslide Recognition and Transferability Using Object-Based Image Analysis in Brazil, Remote Sens., 1–16, 2023.

Schulz, W. H.: Landslides mapped using LIDAR imagery, U.S. Geol. Surv. Open-File Rep., Seattle, W, 1396, 2004.

---

## Author Comment (AC2)

**Inventory mapping of forest-covered landslides using Geographic Object-Based Image Analysis (GEOBIA), Jena region, Germany**

Authors' response

Anonymous Referee #2

The manuscript presents a very interesting and potentially valuable contribution to the NHESS. The incorporation of GEOBIA into landslide mapping represents a notable advancement in this domain. However, several parts of the manuscript require further modifications and improvement before it can be considered for publication in the journal.

**Comment:** Authors should outline the specific landslide features that their method is able to identify. In the current case, it seems that we are discussing structured landslide failures such as translational slides.

**Response:**

> Thank you for the comment. We agree that clarification is helpful, and we have added a brief explanation in the last paragraph (revised manuscript, at line 83) of the Introduction, as we primarily focused on deep-seated (rotational) landslides.

**Comment:** The authors mention that the area of interest has witnessed several landslide events, but without any clarification of the type of movement or the temporal resolution of the events. Are there event-based failures or is there a temporal scale of their occurrence?

**Response:**

> Thank you for pointing this out, and agree that further clarification is needed. In the revised manuscript (lines 105-109), we clarify that the dominant type of movement in the study area consists of deep-seated rotational landslides, which are characteristic of cuesta landscapes with layered sedimentary rocks.
>
> The age of these landslides is not precisely known and cannot be derived directly from the DTM data. However, as explained in the revised manuscript (lines 111–114), based on literature and geomorphological indicators, such as the widespread presence of dense forest cover over many large landslide bodies, we assume they are of Holocene origin with limited recent activity.

According to Achilles et al. (2016), and as now stated in the revised manuscript (lines 109–111), these landslides were probably triggered at the end of the Weichselian glaciation by increased precipitation, glacial meltwater infiltration, and related hydrological changes.

**Comment:** Regarding the manual mapping of landslides, authors should provide clear information on the procedure they followed to map the landslide features. This is crucial for the reader to understand the process of the accuracy assessment in the later stage.

**Response:**

Thanks for this comment. In the revised manuscript, at lines 132-141, we have clarified the criteria used for visual landslide mapping. The landslide inventory map (reference map) was produced through manual visual mapping in ArcMap 10.7, primarily based on traditional and multi-directional hillshade derived from 1m LiDAR–DTM data. This method follows the procedure described by Schulz (2004), which is already cited.

Although hillshade was the only data type directly used to create the inventory, additional land-surface variables (LSVs), such as slope, curvature (plan and profile), topographic openness, topographic position index (TPI) and terrain ruggedness index (TRI), were employed to assist with on-screen interpretation. These LSVs were particularly useful for improving the delineation of landslide boundaries where hillshade alone did not provide sufficient contrast. In most cases, the landslide scarp and body were mapped separately if they could be visually distinguished; however, in a few instances, identification of the scarp was not possible.

**Comment:** The highlighted advantage of this work is the application of the GEOBIA. The process of identifying objects instead of pixels is crucial and it gives the power for semantic labeling and contextual information incorporation. In this case authors should talk and discuss further the parameters for the segmentation phase, such as scale, shape/color, and compactness. More information is needed on the ruleset development and an explanation of the chosen parameters.

**Response:**

Thank you for this insightful comment. We fully agree that a detailed discussion of the segmentation parameters is essential to clarify the methodological robustness and enhance the transparency and transferability of our GEOBIA-based approach. While the original manuscript provided a general overview, we have now substantially expanded this section in the revised version. Specifically, we have included a comprehensive summary of the segmentation and classification parameter, such as **scale (**following trialand-error approach), shape, and compactness (default parameters), at lines 171-176, in the revised manuscript, as well as the corresponding ruleset logic now added in Appendices A1–B2 (see revised manuscript). These additions provide explicit thresholds and decision rules used for the identification of both landslide scarps and bodies, thereby offering a clearer understanding of the classification strategy and its potential for adaptation to other study areas.

Comment: The section on Refinement and accuracy assessment (AA) needs more clarification. I propose to improve it by incorporating more information on how and why the procedure is critical for assessing the performance of the method.

Response:

Thank you for this important and constructive suggestion. We fully agree that the refinement and accuracy assessment steps are critical components of our methodology and required more clarification. Accordingly, we have substantially revised and expanded this section in the revised manuscript to better explain both the rationale and implementation of the refinement procedure. In particular, we now describe the purpose and structure of Stage III (GEOBIA-based refinement) in detail, including how expert knowledge was incorporated through a rule-based approach implemented in eCognition (see lines 178–206 in the revised manuscript). This includes explanation of key object-based features used in the refinement—such as morphometric, geometric, and contextual parameters—and the rationale for their selection.

We also clarified the iterative process of refinement, including how visual inspection, spatial inconsistencies, and accuracy metrics guided the semi-automated adjustments. Specific rules and thresholds used for reclassification (e.g., adjacency, shared boundaries, enclosure) are now explicitly included. Additionally, we noted that landslide scarps and bodies were refined using separate criteria to account for their different spatial characteristics. To ensure transparency and reproducibility, we referenced the full rule set provided in the Appendices (Tables A1–B2), as noted at the end of this section. The updates aim to clearly demonstrate how this refinement phase improved classification performance and why it was essential for reducing false positives and false negatives while maintaining high true positive rates.

We hope these revisions fully address your concerns and provide a clearer, more informative explanation of this critical methodological step.

Comment: The Results section would benefit from a more detailed and thorough presentation. Please provide a deeper interpretation of the findings to enhance clarity and understanding for the reader.

Response:

Thank you for this valuable comment. We fully agree, and in response, we have added two tables in Section 4.2 ("GEOBIA-based landslide modelling results") to provide greater detail and clarity regarding our findings. These tables (Tables 2 and 3 in the revised manuscript) present key morphological parameters for the GEOBIA-based mapping results, including both scarps and bodies, for Models I and II. Brief interpretations of these results have also been incorporated into the revised manuscript at lines 262 and 271. We further modified and improved Figure 6 (from the original manuscript) by splitting it into two figures (Figures 6 and 7) in the revised manuscript. Thus, this improvement provides a clearer presentation of our results as well. We hope that the inclusion of these tables and modification of the figures, along with the explanatory text, improves the clarity and interpretation of our results and addresses your concern as well.

**Comment:** There are several figures that look blurred on the manuscript. Please take a look at them and provide better quality as outputs to enhance the quality of the work.

**Response:**

Thank you for pointing this out. We agree that the figures should be clearly readable across all file types. Accordingly, we have modified the study area figure (Fig. 1), and we have also revised Figures 4, 5, and 7 (renumbered as Fig. 8 in the revised manuscript) to further improve clarity. In the updated figures, landslide scarps and bodies from the inventory and the GEOBIA results are depicted using distinct colors, which we believe improves visual interpretation and facilitates clearer comparison. Additionally, we have split the original Figure 6 into two separate figures (now Figures 6 and 7 in the revised manuscript) and revised them with higher resolution, larger dimensions, and improved layout. We have also updated the figure captions to be more detailed and descriptive. We hope these revisions effectively address your concerns (see the revised manuscript).

---

## Author Comment (AC3)

**Inventory mapping of forest-covered landslides using Geographic Object-Based Image Analysis (GEOBIA), Jena region, Germany**

Authors' response

Community comments (CC):

CC1: Mihai Niculita
Comment: This is an approach for translational landslides, and this should be specified in the title.

Response:

> Please also refer to our response to Reviewer 1, Comment 1. We have added information about the landslides investigated in this study to the Introduction section (revised manuscript, line 83).

Comment: Beside that we will show bellow that the approach is actually not able to predict correctly: we would expect an object for every landslide, in order to be able to validate, but this is not the case so the area based metrics was introduced.

Response:

> This would be an ideal situation, however not a standard in object-oriented analysis. Area-and object-based metrics are more meaningful in evaluating the results of the object-oriented analysis, since they do not only assess the spatial coincidence, but the degree of spatial overlapping as well.

Comment: GEOBIA has potential in landslide research but the presented approach does not progress beyond what was already done in the literature (van den Eckhaut for example); this is shown by the results of stage II. The stage III is nothing more than an approach for (over)fit the landslide data, so its usage outside the study area is questionable. The failure of the segmentation approach is shown by the failure to identify the bodies especially, since their roughness is pretty different than of the surrounding hillslopes as it can be seen in Figure 7. So the big problem remains the segmentation approach which seems to get entire hillsopes rather than the landslides. Also the inventory is questionable: for example in Fig. 7 b the very wide landslide actually is composed on several clear events that should be mapped and considered separatelly. Scarp areas in this context of translational landslides is very hard to be morphometrically segmented.

Response:

1. A thorough examination the Van Den Eeckhaut et al. (2012) study reveals that, while they employed a GEOBIA approach in general, the specific criteria and steps we used differ significantly. For example, in Fig. 5E of their study, they manually created flanks for each landslide in a loop, which is entirely different from our approach. In our study, Stage II is substantially different from their methodology, and Stage III advances further by automating landslide detection in forested areas using high-resolution DEMs, without relying on other data sources.

   Our work takes a step forward by developing a model to map and semi-automatically clean up false positives. As demonstrated in our study, our method minimizes false positives more effectively compared to the previous studies in the same direction (refer to the Discussion section of our study).

2. Thank you for your comment regarding the inventory map, particularly Figure 7b (Fig 8b, revised manuscript). We agree that the large polygon in this figure may in fact represent multiple landslide events (or some secondary landslides). However, due to unclear geomorphological boundaries and possible anthropogenic modifications, it was not possible to confidently delineate the individual events. As a result, we mapped the area as a single landslide to ensure consistent validation, while acknowledging this limitation.

   To address your concern, we have added a paragraph in the Discussion section (at lines 328–333, revised version), where we explain that this issue can be handled through the area-based accuracy assessment approach (see Section 4.3.1). We also suggest potential strategies for dealing with such ambiguous cases in future studies.

**Comment:** The discussions should also point the fact that the proposed approach identify and not necessarily map the landslides. So the method does say there in this object there is a landslide but does not map its borders. Also the validity of the landslide inventory in terms of events should be questioned here. Many landslides are rather compound then single events and this does affect the application of the method.

**Response:**

Many studies on landslide detection using GEOBIA highlight the difficulty of accurately delineating landslide borders, especially compared to manually created inventory maps (Dias et al., 2023). This challenge is influenced by various factors, such as the modification of old landslide boundaries over time by both human and natural processes, and the quality of inventory mapping itself. For further details on the quality of landslide mapping, refer to (Guzzetti et al., 1999, 2012; Santangelo et al., 2010; and more resently

Ardizzone et al., 2023); This issue is also common in geomorphological mapping and other geomorphic features.

However, in our study, we were able to delineate landslide borders, including both scarps and bodies, using Model II. For example, in the central part of the study area (as seen in Fig. 7b), the right-hand landslide is mapped with approximately 90% accuracy compared to the inventory map (pink polygon/body, updated version). This demonstrates that our developed Model MII effectively maps landslides where geomorphological signatures are well-preserved under forest cover, even after hundreds of years.

That said, as shown in Fig. 7, some areas are not fully or accurately mapped. This is partly due to shared roughness between the landslide and the surrounding terrain, and where the GEOBIA merge small portions of true negatives with true positive were removed partly or vice versa and this type of misclassification recorded in other studies (see Knevels et al., 2019; Dias et al., 2023). Our approach prioritizes minimizing false positives while aiming for inventory mapping over a larger area (150 km$^2$) rather than perfect detection of individual landslides. Additionally, we have highlighted areas and examples where the model was unable to detect landslides accurately and these limitations can be addressed in future research steps.

Regarding the transferability of the model (MII), we assume it can be applied to other regions and larger areas. However, as noted in our results (Figures 4 and 5), this depends on the availability of high-resolution DTM data and the clear presentation of landslide features. MII can effectively detect and map landslides (scarps and bodies) when these features are well-represented in the DTM.

We acknowledge that the model (MII) and the ruleset in eCognition may require adaptation for different study areas or other landslide types. Nonetheless, this is the first model to use scaled LSVs for scarps and bodies detection and demonstrate the optimal moving window size for each feature. We also assume, supported by previous studies, that geomorphological features may require different moving window sizes for each LSV, which can vary between features. For instance, in landslides, the window sizes differ between scarps and bodies (see Figure 3 and Table 1, in the manuscript).

In the revised manuscript, we presented the morphological characteristics of the GEOBIA-based results from our models, highlighting clear dissimilarities between scarps and bodies in terms of mean LSV values. This demonstrates the feasibility of detecting them separately. For more details, please refer to Tables 2 and 3 in the revised version. Additionally, Figure 3 and Table 1 illustrate the differences between the default model (MI) and the optimized model (MII) regarding assigned LSVs.

The issue of scale has been discussed in numerous studies, and it remains a central focus of our research as the first study to incorporate both landslide

components. While it would be fundamentally easier to map landslides as single polygons rather than distinguishing and classifying their components, as some studies have done, our approach aims to refine this process.

Our next research steps will aim to generalize the model to a wider area in Germany to evaluate its transferability at a regional scale. We will also explore further adaptations of the model and assess how effectively we can minimize LSVs and streamline the refinement steps. This research marks an initial step in this direction and in advancing landslide mapping the surrounding region and in Germany, however, we assume that different mass movements may require different criteria and parameter.

**Comment.** Lines 42-59 present a sparse review of GEOBIA applications in landslides, without clearly stating the state-of-the-art in this regard; since the approach is considered to be an advance, it should be framed better.

**Response:**

While many studies have utilized GEOBIA to detect landslides, we provided only a brief review, beginning broadly before narrowing the focus. As our study specifically addresses the application of GEOBIA for landslides in forested areas, and more specifically, for identifying old landslides, we have included the most relevant studies within this context. This is why we transitioned from general landslide detection using GEOBIA to the challenges of detecting old landslides based solely on DTM data.

Additionally, we aimed to address the issue of scale, which is a central aspect of our study (see Model II). For this reason, we provided an overview of scale-related challenges, from landform classification to landslide-specific applications. Initially, we even considered beginning directly with the topic of old landslides in vegetated areas using GEOBIA (relying exclusively on DEM data) and focusing solely on the issue of scale. Therefore, while we acknowledge the possibility of including additional studies, we respectfully consider the current review to be appropriately scoped for the objectives of this manuscript.

**References**

Ardizzone, F., Bucci, F., Cardinali, M., Fiorucci, F., Pisano, L., Santangelo, M., and Zumpano, V.: Geomorphological landslide inventory map of the Daunia Apennines, southern Italy, Earth Syst. Sci. Data, 15, 753–767, https://doi.org/10.5194/essd-15-753-2023, 2023.

Dias, H. C., Hölbling, D., and Grohmann,  and C. H.: Rainfall-Induced Shallow Landslide Recognition and Transferability Using Object-Based Image Analysis in Brazil, Remote

Sens., 1–16, 2023.

Van Den Eeckhaut, M., Kerle, N., Poesen, J., and Hervás, J.: Object-oriented identification of forested landslides with derivatives of single pulse LiDAR data, Geomorphology, 173–174, 30–42, https://doi.org/10.1016/j.geomorph.2012.05.024, 2012.

Guzzetti, F., Carrara, A., Cardinali, M., and Reichenbach, P.: Landslide hazard evaluation: a review of current techniques and their application in a multi-scale study, Central Italy, Geomorphology, 31, 181–216, 1999.

Guzzetti, F., Cesare, A., Cardinali, M., Fiorucci, F., Santangelo, M., and Chang, K.: Landslide inventory maps : New tools for an old problem, Earth Sci. Rev., 112, 42–66, https://doi.org/10.1016/j.earscirev.2012.02.001, 2012.

Knevels, R., Petschko, H., Leopold, P., and Brenning, A.: Geographic object-based image analysis for automated landslide detection using open source GIS software, ISPRS Int. J. Geo-Information, 8, https://doi.org/10.3390/ijgi8120551, 2019.

Santangelo, M., Cardinali, M., Rossi, M., Mondini, A. C., and Guzzetti, F.: Remote landslide mapping using a laser rangefinder binocular and GPS, Nat. Hazards Earth Syst. Sci., 10, 2539–2546, https://doi.org/10.5194/nhess-10-2539-2010, 2010.